# Orchestration of microtubules and the actin cytoskeleton in trichome cell shape determination by a plant-unique kinesin

**Juan Tian[1], Libo Han[1], Zhidi Feng[1], Guangda Wang[1,2], Weiwei Liu[1,2], Yinping Ma[1,2], Yanjun Yu[1], Zhaosheng Kong[1]\***

[1]State Key Laboratory of Plant Genomics, Institute of Microbiology, Chinese Academy of Sciences, Beijing, China; [2]University of Chinese Academy of Sciences, Beijing, China

**Abstract** Microtubules (MTs) and actin filaments (F-actin) function cooperatively to regulate plant cell morphogenesis. However, the mechanisms underlying the crosstalk between these two cytoskeletal systems, particularly in cell shape control, remain largely unknown. In this study, we show that introduction of the MyTH4-FERM tandem into KCBP (kinesin-like calmodulin-binding protein) during evolution conferred novel functions. The MyTH4 domain and the FERM domain in the N-terminal tail of KCBP physically bind to MTs and F-actin, respectively. During trichome morphogenesis, KCBP distributes in a specific cortical gradient and concentrates at the branching sites and the apexes of elongating branches, which lack MTs but have cortical F-actin. Further, live-cell imaging and genetic analyses revealed that KCBP acts as a hub integrating MTs and actin filaments to assemble the required cytoskeletal configuration for the unique, polarized diffuse growth pattern during trichome cell morphogenesis. Our findings provide significant insights into the mechanisms underlying cytoskeletal regulation of cell shape determination.

**\*For correspondence:** zskong@im.ac.cn

**Competing interests:** The authors declare that no competing interests exist.

## Introduction

Plant cells assume an amazing diversity of cell shapes that enable these cells to execute unique physiological functions, and the study of plant cell shape determination has remained an intriguing part of plant biology (*Smith and Oppenheimer, 2005*; *Szymanski, 2009*). The plant cytoskeletal system, composed of microtubules (MTs) and actin filaments (F-actin), plays a central role in cell morphogenesis in both tip-growing and diffuse-growing cell types. In particular, the cortical MTs play pivotal roles in plant cell growth and directionality, mainly by orienting the deposition of nascent cellulose microfibrils during biosynthesis of the cell wall (*Paradez et al., 2006*). F-actin plays central roles in polarized cell elongation, mainly by regulating intracellular transport (*Hussey et al., 2006*). However, despite emerging evidence that cortical MTs and F-actin coordinately regulate plant cell morphogenesis (*Petrasek and Schwarzerova, 2009*; *Sampathkumar et al., 2011*; *Sambade et al., 2014*), the molecular mechanisms underlying the crosstalk between these two cytoskeletal systems remain largely unknown.

The *Arabidopsis thaliana* leaf trichome, a single cell bearing three or four branches on top of a stalk, has long been a model system for investigating the role of the cytoskeleton in defining plant cell shape (*Smith, 2003*; *Szymanski, 2009*). The trichome phenotype indicates the cytoskeletal homeostasis inside the plant cell. For example, cortical MTs mainly affect initiation of trichome branches, and disruption of genes encoding MT-associated proteins usually produces trichomes with abnormal numbers of branches (*Oppenheimer et al., 1997*; *Krishnakumar and Oppenheimer, 1999*; *Burk et al., 2001*; *Buschmann et al., 2009*; *Nakamura and Hashimoto, 2009*; *Kong et al., 2010*). By contrast, actin filaments mainly affect elongation of trichomes, and disruption of genes encoding actin-related proteins, such as components of the ARP2/3 actin nucleation complex and the upstream

**elife digest** Within a cell, a structure called the cytoskeleton provides a scaffold that supports the cell's shape. In both plant and animal cells, this scaffold is largely made of tube-like structures called microtubules and a web of filaments made of a protein called actin.

In a plant called *Arabidopsis thaliana*, specialized hair-like cells usually protrude from the surface of leaves. These cells—called trichomes—are widely used to study how the cytoskeleton influences plant cell shape. Microtubules are required to regulate the number of branches these trichomes have. The actin filaments control the length of the trichomes, but it is not known how the two elements of the cytoskeleton interact to determine the overall shape of the trichomes.

One protein that might help to co-ordinate microtubules and actin filaments is called kinesin-like calmodulin-binding protein (or KCBP for short). This protein is a type of motor protein known as a kinesin and can move along microtubules, but it also contains a section called the MyTH4-FERM domain, which is found in another type of motor protein. When KCBP is removed from *A. thaliana*, the leaf trichomes are shorter and blunter than normal and have fewer branches. Here, Tian et al. investigated the role of KCBP in trichome development using a combination of genetic and microscopy techniques.

The experiments show that the MyTH4-FERM domain of KCBP allows the protein to bind to both actin filaments and microtubules. During trichome formation, KCBP associates with the microtubules that are attached to the membrane that surrounds the cell. Particularly, high levels of the protein accumulate at the sites where branches will form and at the tips of developing branches. These branch tips have few microtubules but have many actin filaments.

Further experiments revealed that KCBP acts as a hub that brings together microtubules and actin filaments to modulate the cytoskeleton during trichome formation. Tian et al.'s findings provide new insights into how the cytoskeleton influences plant cell shape. A future challenge is to understand how KCBP prevents microtubules forming at the tips of growing branches during trichome formation.

WAVE/SCAR regulatory complex (W/SRC), usually results in impaired trichome elongation but with normal branch number (*Mathur et al., 2003*; *Basu et al., 2004*; *Deeks et al., 2004*; *El-Assal Sel et al., 2004*; *El-Din El-Assal et al., 2004*; *Li et al., 2004*; *Szymanski, 2005*; *Zhang et al., 2005a*; *Djakovic et al., 2006*; *Le et al., 2006*). Therefore, theoretically, mutation in a gene that participates in both processes would lead to trichome defects in both branching and elongation and would assist in the elucidation of the crosstalk between the MTs and the F-actin. However, such a central player, which directly links and integrates MTs and F-actin and further establishes the required cytoskeletal systems for trichome cell shaping, remains to be identified.

KCBP (kinesin-like calmodulin-binding protein) occurs solely within the plant kingdom, and uniquely in the kinesin superfamily. KCBP has a C-terminal motor head, beyond which locates a calmodulin-binding domain (CBD) at the extreme end of the C-terminus; besides KCBP, the CBD is only found in the sea urchin kinesin KinC. KCBP also contains a MyTH4-FERM tandem, which only occurs in several myosin families outside the plant kingdom, in its tail region at the N-terminus (NT); thus, KCBP has long been regarded as a chimera of kinesin with a myosin (*Abdel-Ghany et al., 2005*). Using biotinylated calmodulin as a probe, KCBP was firstly isolated as a novel calmodulin-binding protein with a kinesin motor domain (*Reddy et al., 1996*). In vitro biochemical assays showed that the CBD of KCBP binds calcium/calmodulin, which negatively regulates KCBP motor activity, and that the N-terminal tail, including the MyTH4-FERM domain, could co-sediment with MTs (*Narasimhulu and Reddy, 1998*). Interestingly, genetic analysis of the *zwichel* (*zwi*) mutants, which have a shortened stalk and only two branches, revealed that the *ZWICHEL* (*ZWI*) gene encodes the KCBP protein (*Hülskamp et al., 1994*; *Oppenheimer et al., 1997*). KCBP was detected in mitotic MT arrays such as the preprophase band, the spindle, and the phragmoplast in various plants, indicating that it may play a role in cell division (*Bowser and Reddy, 1997*; *Smirnova et al., 1998*; *Preuss et al., 2003*; *Buschmann et al., 2015*). Nevertheless, loss of function in *KCBP* only produces a noticeable phenotype in trichomes, which are less-branched and contain shortened stalks and swollen, stunted branches (*Oppenheimer et al., 1997*), providing a hint of clue that KCBP possibly plays the role involving in both cortical MTs and F-actin.

However, whether KCBP localizes to cortical MTs in trichome cells is still an open question, cellular basis of defects in *kcbp* trichomes needs to be examined, direct evidence linking KCBP and F-actin is currently missing, and the role of the mysterious MyTH4-FERM domain remains to be unraveled.

Here, we show that the N-terminal tail comprising the MyTH4 domain strongly binds to MTs, and the FERM domain physically binds to F-actin. During trichome morphogenesis, KCBP localizes to cortical MTs, distributes in a specific gradient, and is concentrated at the branching sites and the apexes of elongating branches. KCBP orchestrates the MT-actin interplay and is required to assemble the specific cytoskeletal configuration for trichome branching, elongation, and tip sharpening. Our findings build the direct link between the MT-based KCBP motor and the actin cytoskeleton, unravel the cellular basis controlling trichome cell shaping, and provide significant insights into the mechanisms of cytoskeletal regulation of cell shape determination.

## Results

### KCBP localizes to cortical MTs in the non-processive mode

To determine whether KCBP localizes to cortical MTs, we performed live-cell imaging using a functional, GFP-tagged KCBP fusion under the control of its endogenous regulatory elements; this construct fully rescued the typical *zwichel* trichome defects in the *kcbp-1* (Salk_031704 in the *Arabidopsis* Biological Resource Center; see 'Materials and methods') mutant (*Figure 1—figure supplement 1A*) (*Humphrey et al., 2015*), which was also designated as *zwiA* (N531704 in the Nottingham *Arabidopsis* Stock Centre; see 'Materials and methods') (*Buschmann et al., 2015*). Notably, we found that GFP-KCBP localizes to puncta along cortical MTs in both pavement cells and hypocotyl cells (*Figure 1A*, *Figure 1—figure supplement 2B*, *Video 1*, *Video 2*). Most GFP-KCBP proteins dwell on or move along cortical MTs as discrete particles for a short time (15.9 s, n = 647); only a small portion of GFP-KCBP particles remain immobilized on MTs for longer times (*Video 1*, *Video 2*). Kymograph analysis further confirmed the short-lived and poorly processive nature of KCBP movement (*Figure 1B*, *Figure 1—figure supplement 2C*), which is the characteristic of non-processive motors of the kinesin-14 subfamily (*Fink et al., 2009*). We then quantified the distributions of the velocity and the run-length observed for GFP-KCBP. The distributions indicated that, upon fitting to an exponential line, the motor moved at an average velocity of 0.68 µm/min (n = 647) (*Figure 1C*, *Figure 1—source data 1*), and at an average run length of 0.18 µm (n = 647) (*Figure 1D*, *Figure 1—source data 1*). Taken together, these results indicated that KCBP likely acts as a non-processive motor to facilitate crosslinking or bundling during MT organization.

### Enrichment of KCBP at the trichome-branching sites and in the tip region of elongating branches

To gain further insights into the role of KCBP in regulating trichome morphogenesis, we examined the spatio-temporal dynamics of GFP-KCBP during trichome morphogenesis. Indeed, we observed that KCBP largely colocalizes with cortical MTs in developing trichomes (*Figure 2—figure supplement 1*). Intriguingly, KCBP forms a gradient in the elongating trichome branch, with the highest density at the extreme apex, which is devoid of MTs (*Figure 2—figure supplement 1*). Because the signal of mCherry-labeled MTs is much weaker than that of GFP and undergoes rapid photobleaching, it is extremely hard to detect in trichomes due to their three-dimension geometry; therefore, to ensure an accurate comparison, we further observed KCBP and MTs using the GFP-KCBP-expressing line and the GFP-TUB6-expressing line, respectively. We observed punctate KCBP localization in the cortical gradient in developing trichomes, with the highest density at the extreme apex (*Figure 2A,D,G*; *Figure 2—figure supplement 2A*; *Video 3*, *Video 4*). In GFP-TUB6-expressing trichomes at identical developmental stages, we observed transversely aligned MT arrays (transverse MT rings) in a similar gradient with higher MT density towards the branch tip, but forming a MT-depleted zone at the extreme apex (*Figure 2B,E,H*; *Figure 2—figure supplement 2B*; *Video 3*). In consistent with the result from the double-labeling imaging (*Figure 2—figure supplement 1*), KCBP largely colocalizes with cortical MTs along the trichome branch, but shows highest levels in the MT-depleted zone at the branch apex (*Figure 2A,B,G,H*; *Video 3*, *Video 4*). We also found a similar distribution at the branching site where KCBP accumulates, but which has a relatively sparse MT network (*Figure 2A,B,D,E*; *Video 3*). To further validate the presence of the MT-depleted zone, we used GCP2 (*Liu et al., 2014*), a component of the MT nucleation complex, as a control. Interestingly, we observed that, similar to MTs,

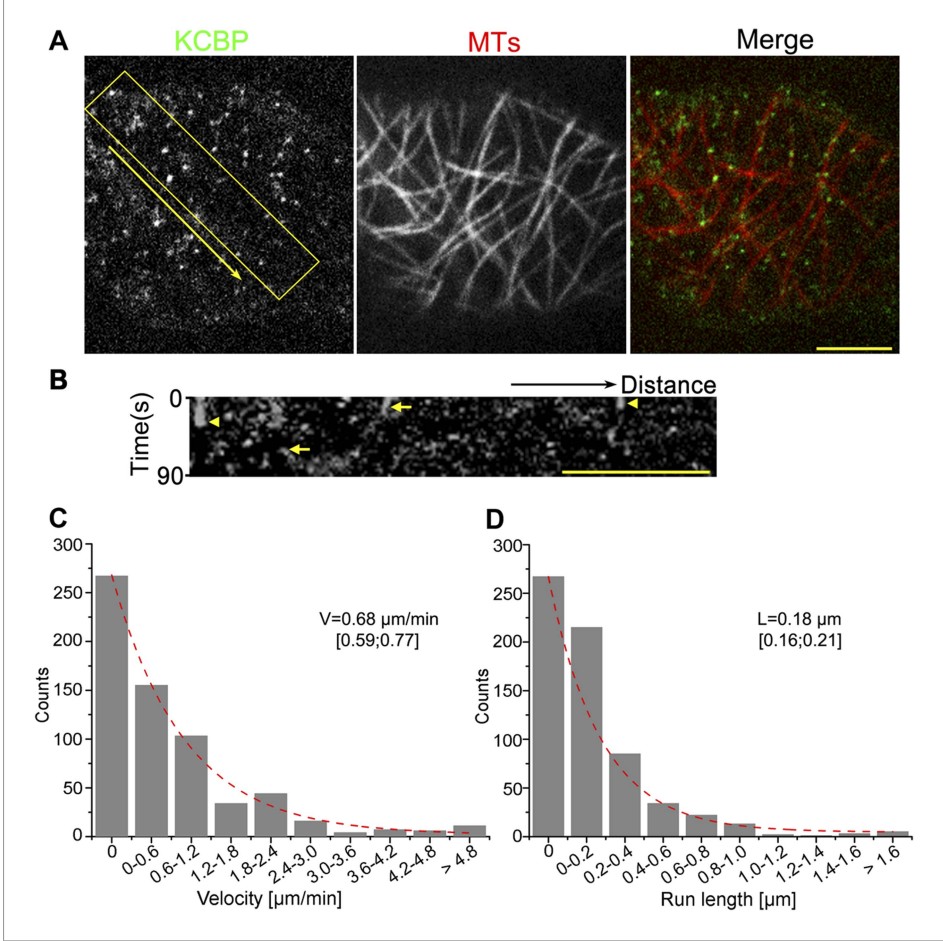

**Figure 1**. KCBP colocalizes with cortical MTs in vivo. (**A**) GFP-labeled kinesin-like calmodulin-binding protein (KCBP) localizes along cortical microtubules (MTs) (mCherry-TUB6) in a punctate pattern in *Arabidopsis* epidermal pavement cells. The yellow box highlights the area used to generate the kymograph and the yellow arrow marks the direction for the kymographic analysis in (**B**). See also *Video 1*. (**B**) Kymograph showing the dynamicity of GFP-KCBP particles. The bright dots indicate transient appearance of most GFP-KCBP particles and the linear tracks indicate the motility of GFP-KCBP particles either dwelling on (marked by arrowheads) or moving along MTs (marked by arrows) for a short time. (**C**, **D**) Distribution of the velocity (**C**) and the run length (**D**) of GFP-KCBP moving along cortical MTs. The mean values are shown with standard deviations and examined sample sizes. Dashed lines represent the trends derived from exponential fits. Scale bars, 5 μm.

The following source data and figure supplements are available for figure 1:

**Source data 1**. Distribution of the velocity and the run length of GFP-KCBP moving along cortical MTs.

**Figure supplement 1**. Genetic identification of the *kcbp-1/zwiA* mutant.

**Figure supplement 2**. KCBP colocalizes with cortical MTs in *Arabidopsis* hypocotyl cells.

GCP2 shows a tip-directed gradient, but with a GCP2-depleted zone at the extreme apexes of elongating branches (*Figure 2—figure supplement 3*; *Video 5*).

The enrichment of KCBP at the MT-depleted zone induced us to compare the distribution of KCBP with the actin cytoskeleton in developing trichomes at identical developmental stages. We observed that F-actin forms thick bundles parallel to the long axis of elongating branches and extend from the stalk to the region near branch tips, but also form a cap with fine, cortical F-actin mainly in a transverse pattern in the tip region (*Figure 2C,F,I*; *Video 3*). Notably, the transverse cortical F-actin cap occupies

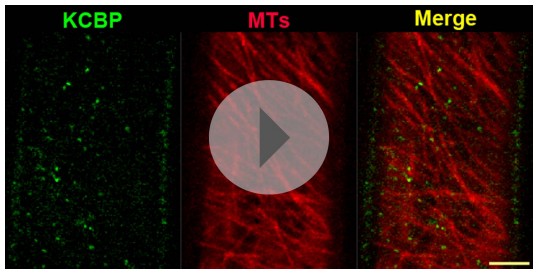

**Video 1.** Localization and dynamicity of kinesin-like calmodulin-binding protein (KCBP) on cortical microtubules (MTs) in *Arabidopsis* epidermal pavement cells. Images were obtained at 3-s intervals. A total of 30 time lapse images were applied to make the video. Scale bar, 5 μm.

a larger area than the MT-depleted zone, but coincides with the area where KCBP accumulates (*Figure 2A–I*).

Together, our observations suggest that the specific distribution of KCBP is tightly associated with the assembly of the required cytoskeletal organization for trichome branching and elongation.

## The transverse MT rings and the MT-depleted zone are disorganized in *kcbp-1* trichomes

To further reveal the cellular mechanisms by which KCBP affects trichome morphogenesis, we monitored MT organization at various stages of trichome development in *kcbp-1* mutants and the wild-type control. The wild-type *Arabidopsis* leaf trichome initiates from a bulging protodermal cell and then grows out into a cylindrical cell, which exhibits random organization of the MT array (*Figure 3A*). The random MT network shifts into a transverse MT array (rings) encircling the elongating cylindrical cell (*Figure 3B*). Slightly later, an area near the tip bulges out and gradually forms the primary branch (*Figure 3B–D*). Most trichomes undergo two consecutive branching events to form the typical three-branch pattern (*Figure 3E,F*, *Figure 3—figure supplement 1A*). During trichome branch elongation, cortical MTs also shift to transverse rings with the typical tip-directed MT density gradient (*Figure 3C–F*). Intriguingly, a MT-depleted zone occurs at the extreme apex of the elongating branch (*Figure 3D*; *Video 6*). At the maturation stage, fully elongated trichome branches form a fine and pointed tip, and cortical MTs reorient to an oblique or longitudinal configuration (*Figure 3—figure supplement 1A*).

By contrast, *kcbp-1* trichomes initiate normally, but form a shorter and rounder tubular cell with a randomly oriented MT network (*Figure 3G*). Subsequently, the primary branching event takes place from the severely reduced stalk (*Figure 3H,I*), but the secondary branching does not occur, yielding a two-branch trichome (*Figure 3J,K*, *Figure 3—figure supplement 1B*). Eventually, one branch elongates and forms a tip that is not as pointed as the wild type, whereas the growth of the other branch is impaired, producing a swollen and blunt tip (*Figure 3K*, *Figure 3—figure supplement 1B*). Notably, cortical MT organization in *kcbp-1* trichomes is distinct from that in the wild type. During branching and elongation of the primary branch of *kcbp-1* trichomes, formation of the transverse MT rings that encircle the incipient branch is impaired, and the MT-depleted zone is never observed (*Figure 3H,I*; *Video 7*). In the normally elongating branch (stem), the transverse MT rings are seen in a relatively loose distribution, and the MT-depleted zone is not well defined (*Figure 3I–K*; *Video 7*).

Together, these observations indicated that KCBP is required to assemble the transverse MT rings encircling the elongating branch and formation of the MT-depleted zone, and that loss of *KCBP* function results in impairment of trichome branch initiation and elongation, and branch tip sharpening.

## Parallel cytoplasmic actin cables along the growth axis and the transverse cortical cap at the branch apex are disrupted in *kcbp-1* trichomes

The coincidence of KCBP with the cortical actin network and the exclusion of MTs at the branch apex prompted us to examine whether the organization of F-actin is aberrant during the morphogenesis of *kcbp-1* trichomes. We therefore monitored global actin cytoskeleton changes during trichome development (*Figure 4A–H*). Strikingly, the parallel cytoplasmic

**Video 2.** Localization and dynamicity of KCBP on cortical MTs in *Arabidopsis* hypocotyl cells. Images were obtained at 3-s intervals. A total of 35 time lapse images were applied to make the video. Scale bar, 5 μm.

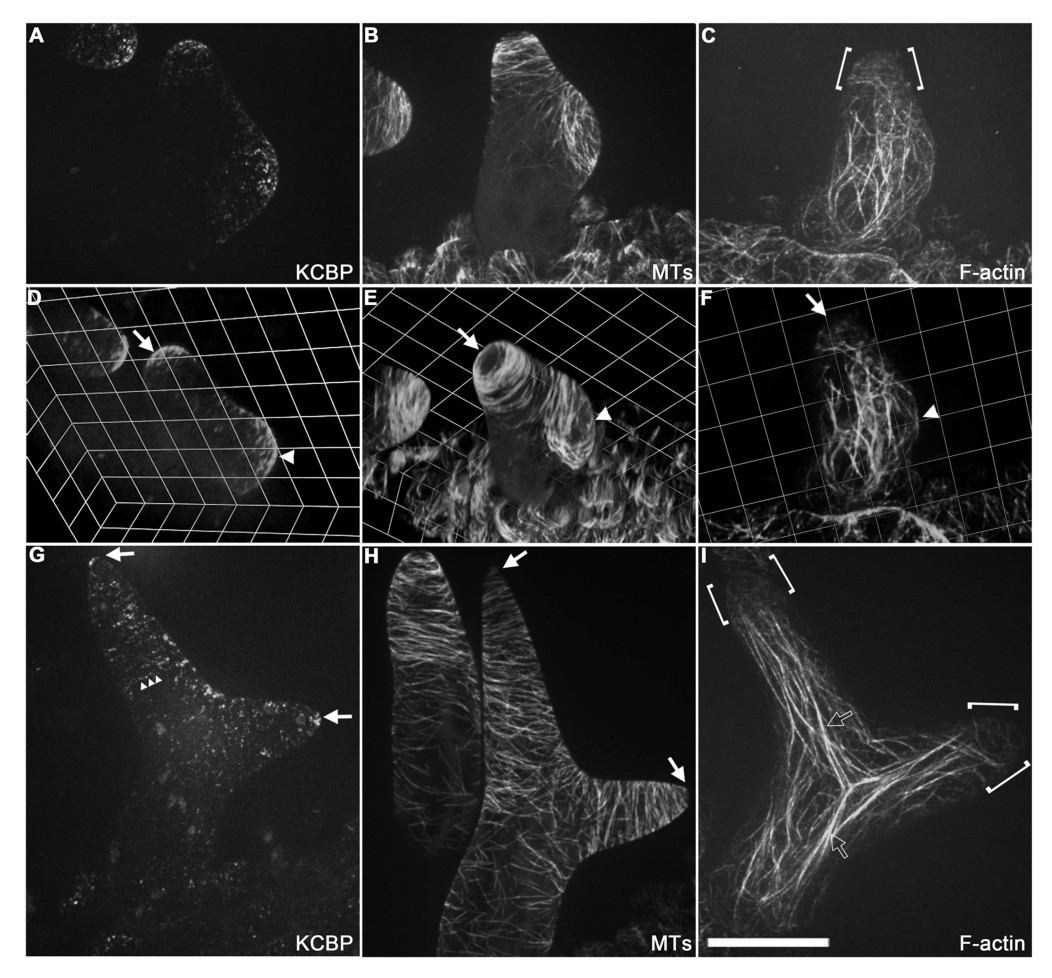

**Figure 2**. Spatio-temporal distribution of GFP-KCBP in developing wild-type trichomes. (**A–F**) Localization of KCBP and spatial organization of the cytoskeleton in stage 2/3 trichomes. GFP-KCBP particles form a cortical gradient with the highest expression at the branching site and the tip region of the main stem (**A**). The arrow and the arrowhead highlight the concentration of GFP-KCBP at the extreme apex of the main stem and the apical region of the incipient primary branch by three-dimension (3-D) reconstruction, respectively (**D**). Transverse MT arrays form rings encircling the elongating main stem, but leave a MT-depleted zone at the extreme apex. The incipient primary branch is also encircled by transverse MT rings, with the apex colonized by sparse MT meshworks (**B**). The arrow and the arrowhead in (**E**) highlight the 3-D reconstruction of the MT-depleted zone at the extreme apex of the main stem and the sparse MT meshworks at the apical region of the incipient primary branch, respectively. Cytoplasmic cables extend along the growth axis of the main stem, from the base to the tip region where they form a fine, cortical F-actin cap mainly in a transverse pattern, highlighted by the brackets (**C**). The arrow and the arrowhead in (**F**) highlight the 3-D reconstruction of the F-actin cap at the tip region of the main stem and actin bundles at the apical region of the incipient primary branch, respectively. See also *Video 3*. (**G–I**) Localization of KCBP and spatial organization of the cytoskeleton in stage 3/4 trichomes. KCBP forms a transverse cortical punctate pattern, with the highest amounts (indicated by arrows) at the extreme apex of the elongating branches. The tandem arrowhead highlights GFP-KCBP particles in the transverse linear pattern (**G**). Transverse MT rings display a tip-directed density gradient, but with a MT-depleted zone (indicated by arrows) at the extreme apex of the elongating branches (**H**). Cytoplasmic actin cables (indicated by open arrows) extend along the growth axis of elongating branches and reach near the tip, where there is a fine, transversely aligned F-actin cap (highlighted by brackets) (**I**). See also *Video 4*. The maximum z-projection of image stacks at 0.2-µm intervals was applied to all figures. Scale bars, 20 µm. One grid unit in (**D–F**) indicates 7.31 µm.

The following figure supplements are available for figure 2:

**Figure supplement 1**. Colocalization of KCBP with cortical MTs in stage 2 trichomes.

*Figure 2. continued on next page*

*Figure 2. Continued*

**Figure supplement 2**. Localization of KCBP and spatial organization of MTs in stage 2/3 wild-type trichomes.

**Figure supplement 3**. Localization of GCP2 in stage 2/3 wild-type trichomes.

actin cables, which extend along the long axis to the branch tip in wild-type trichomes (*Figure 4B–D*), are disorganized in *kcbp-1* trichomes, which instead have a curly, intertwined meshwork of thick actin bundles (*Figure 4E–H*). Moreover, the transverse cortical F-actin cap, which is observed at the branch tip of elongating wild-type trichomes (*Figure 4C*), is disrupted in *kcbp-1* trichomes (*Figure 4G,H*).

Taken together, these findings implicate KCBP in the assembly of parallel cytoplasmic actin cables along the growth axis and the transverse cortical F-actin cap at the branch apex, which are likely required for polarized branch elongation accompanied by tip sharpening, and further suggest that KCBP plays a critical role in regulating the MT-actin interaction.

## The N-terminal tail of KCBP is essential for trichome morphogenesis

To gain genetic insights into the in vivo functions of the individual domains of KCBP, we performed genetic complementation tests using genomic constructs with various domain truncations (*Figure 5A,B*). As expected, the construct without the motor domain could not rescue the trichome morphology of the *kcbp-1* mutant (*Figure 5B*), whereas the construct without the C-terminal CBD domain could perfectly rescue the *kcbp-1* trichome phenotype (*Figure 5B*), clearly indicating that the CBD domain is dispensable during trichome development. Furthermore, we found that constructs with either the NT truncation or the MyTH4 truncation could rescue the *kcbp-1* trichome morphology, but the construct with a truncation of both NT and MyTH4 could not rescue the *kcbp-1* trichome phenotype (*Figure 5B*). In addition, constructs with either truncation of the FERM domain or the (MyTH4+ FERM) domains could rescue the *kcbp-1* trichome morphology, but the construct containing a truncation of the whole (NT+MyTH4+FERM) could not rescue the *kcbp-1* trichome morphology (*Figure 5B*). Importantly, these findings indicated that the NT and the MyTH4 domains are essential for trichome cell shape determination.

## Rigor-KCBP exerts specific dominant-negative effects, genetically supporting the interaction between KCBP and the actin cytoskeleton

In addition to the truncation tests, we introduced a point mutation into *KCBP* that produced a 'rigor KCBP' (T982N) that is defective in ATP (Adenosine 5'-triphosphate) hydrolysis, which is required to perform force-generating tasks (*Figure 6A*). Strikingly, the rigor-KCBP exerted specific dominant-negative effects that resulted in more-severe trichome phenotypes (*Figure 6D,G–N*), resembling a class of trichome mutants with defects in the actin cytoskeleton (*Basu et al., 2004, 2005*; *Li et al., 2004*; *Zhang et al., 2005a*). Most trichomes in plants expressing rigor-KCBP contained two branches but had an extremely twisted and swollen branch (*Figure 6G–I*), and some of them had extremely elongated branches, more than two times as long as the wild-type control (*Figure 6M,N*). Interestingly, a small fraction of trichomes formed three branches but still contained one or two swollen and twisted branches (*Figure 6J–L*).

Moreover, we observed more-severe abnormalities in the organization of both cortical MTs and F-actin in rigor-KCBP trichomes when compared to the wild-type control (*Figure 6—figure supplement 1*, *Figure 6—figure supplement 2*). In addition, the typical, two-branched trichomes in *zwichel* mutants are aligned in a nearly parallel pattern on leaves, with the elongated and pointed branch toward the proximal part and another shortened and blunt-ended branch toward the distal part (*Figure 6C*) (*Hülskamp et al., 1994*; *Oppenheimer et al., 1997*). Notably, in rigor-KCBP trichomes, the elongated and

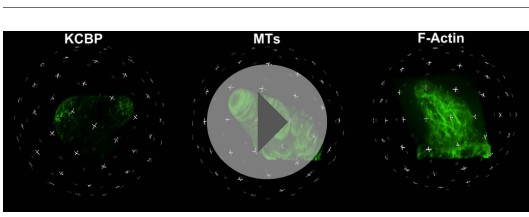

**Video 3.** Spatio-temporal distribution of GFP-KCBP, MTs, and actin filaments is highlighted by 3-D re-constitution in stage 2/3 trichomes.

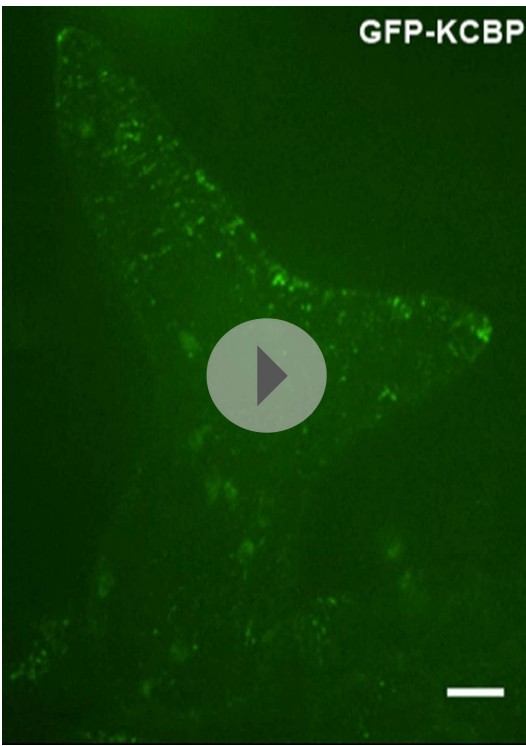

GFP-KCBP

**Video 4.** The spatio-temporal dynamics and distribution of GFP-KCBP in developing trichomes. Images were obtained at 3-s intervals. A total of 8 time lapse images were applied to make the video. Scale bar, 5 μm.

pointed branch toward the proximal part keeps unchanged, and the extremely swollen and twisting specifically takes place in another branch, which derives from the shortened and blunt-ended branch of the *kcbp-1* background (*Figure 6D*). Collectively, these findings genetically supported the idea that KCBP participates in the organization of the actin cytoskeleton during trichome branching and elongation.

## KCBP physically binds to both MTs and F-actin and assembles the required cytoskeletal configuration for trichome cell shaping

To unravel the molecular and cellular basis of the essential role of the N-terminal tail of KCBP and its actin-related function, we purified various truncated versions of motorless KCBP tagged with GFP (*Figure 7—figure supplement 1*) and mapped the specific domain that may physically bind to MTs or F-actin. Single-molecule imaging showed that either the NT (amino acids 1–115) or the MyTH4 domain could bind to MTs, but the FERM domain did not bind MTs; the (NT+MyTH4) region and the (NT+MyTH4+FERM) region showed similar MT-binding activity to that of the single NT or MyTH4 domain, whereas the motorless KCBP (NT+MyTH4+FERM+Coiled Coil) exhibited enhanced MT-binding activity and promoted the formation of MT bundles (*Figure 7A*). Furthermore, the FERM domain directly bound to F-actin; the (NT+MyTH4+FERM) region showed similar F-actin-binding activity as the single FERM domain, whereas the motorless KCBP (NT+MyTH4+FERM+Coiled Coil) exhibited enhanced F-actin-binding and bundling activity (*Figure 7B*).

Taken together, these results demonstrated that KCBP contains a second MT-binding domain spanning the NT and the MyTH4 region, and can also bind to F-actin via the FERM domain, co-ordinating the interplay between MTs and F-actin. We propose that, during trichome morphogenesis, KCBP forms a cortical gradient and highly accumulates at trichome-branching sites and apical regions of elongating branches, thus, integrating MTs and F-actin to assemble the required cytoskeletal configuration for trichome branching, elongation, and tip sharpening (see the model in *Figure 7C*).

## Discussion

In this study, our findings provide conceptual advances both in understanding the role of KCBP in regulating the interaction between cortical MTs and F-actin during trichome morphogenesis and in elucidating the cellular basis of trichome cell shape determination.

GCP2-3×GFP

**Video 5.** Spatio-temporal distribution of GCP2-3XGFP is highlighted by 3-D reconstitution in stage 2/3 trichomes.

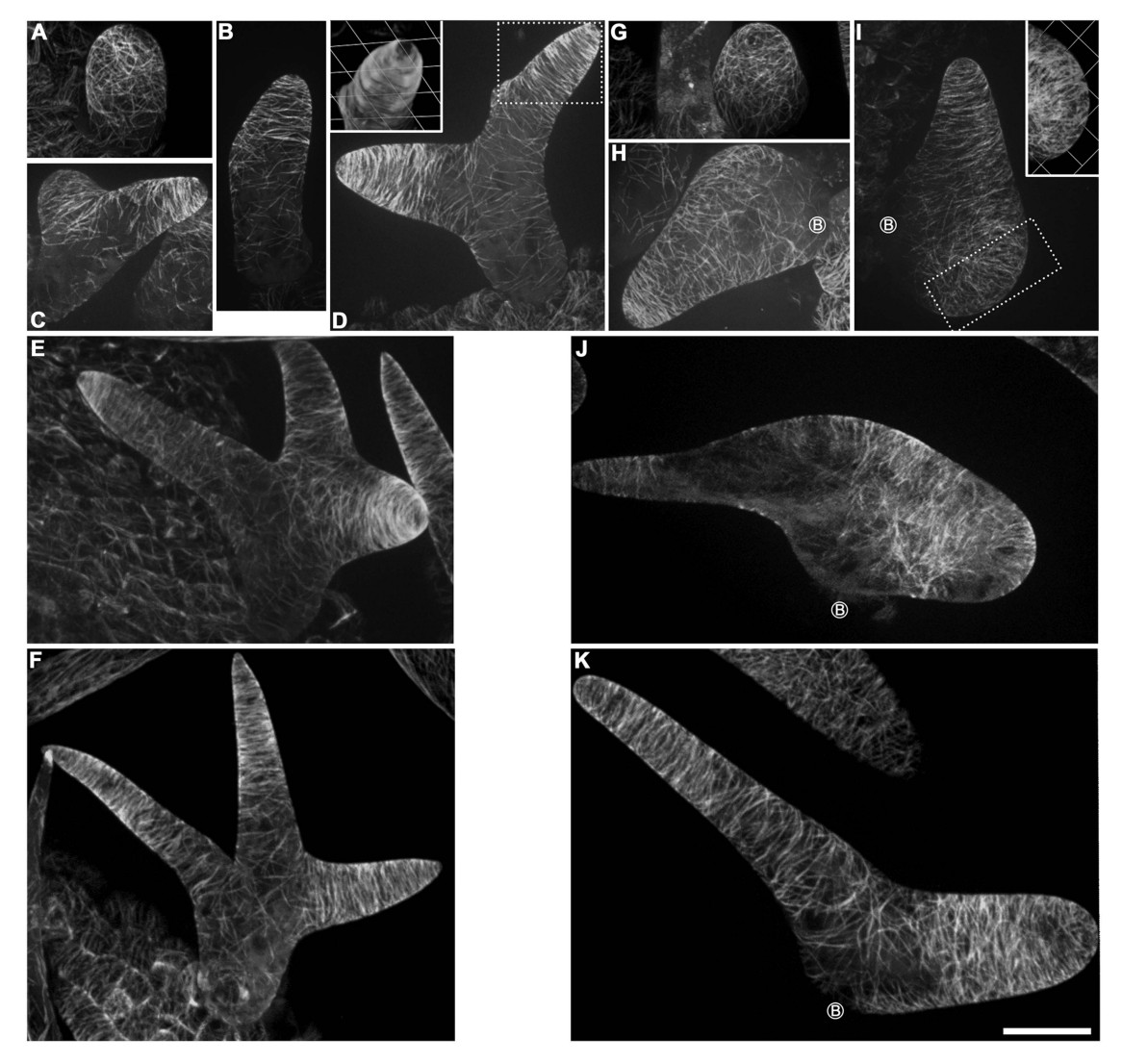

**Figure 3**. Abnormal MT organization in *kcbp-1* trichomes. (**A–F**) Spatio-temporal MT organization in wild-type trichomes during development. The stage 1 trichomes exhibit random MT networks (**A**). In stage 2/3 trichomes, the random MT network shifts into transverse MT rings encircling the elongating main stem and the incipient primary branch (**B**, **C**). In stage 3/4/5 trichomes (**D–F**), cortical MTs rings encircle the elongating branches but leave a MT-depleted zone at the extreme apex, which is highlighted by 3-D reconstruction (the inset in **D**) of the area outlined by the dotted box. See also *Video 6*. (**G–K**) Spatio-temporal MT organization in *kcbp-1* trichomes during development. The stage 1 *kcbp-1* trichomes form shorter and rounder tubular cells with random MT networks (**G**). In stage 2/3 trichomes, formation of the transverse MT rings that encircle the incipient primary branch is impaired (**H**, **I**). The 3-D reconstruction (the inset in **I**) of the incipient primary branch tip (outlined by the dotted box) shows that the MT-depleted zone is not well defined. In stage 3/4/5 trichomes (**J**, **K**), transverse MT rings are present in a relatively loose distribution. The circled B indicates the base of trichome stalks. See also *Video 7*. The maximum z-projection of image stacks at 0.2-µm intervals was applied to all figures. One grid unit in the inset of (**D**, **I**) indicates 7.31 µm. Scale bars, 20 µm.

The following figure supplement is available for figure 3:

**Figure supplement 1**. Cortical MT organization in wild-type and *kcbp-1* mature trichomes.

## KCBP orchestrates the interplay between cortical MTs and F-actin

Due to its unique nature, KCBP is one of the most studied kinesins within the plant kingdom; numerous studies have investigated its biochemical characteristics and potential regulatory roles in cell division. However, since *KCBP* was cloned and linked to the trichome phenotype of the *zwichel*

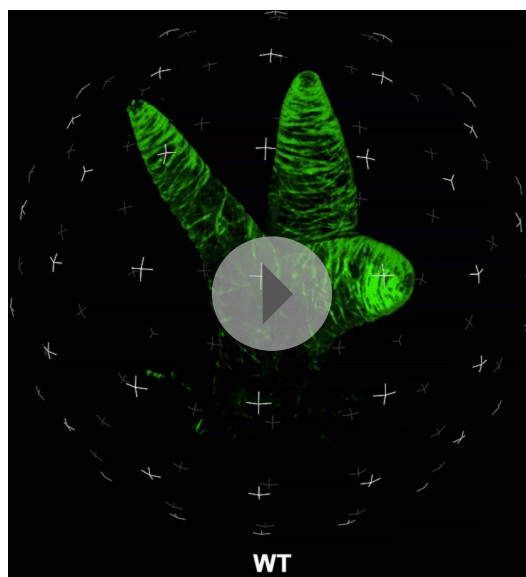

**Video 6.** The 3-D reconstructed cortical MT configuration in stage 3/4 wild-type trichomes.

mutation in *Arabidopsis*, in-depth understanding the in vivo function of KCBP, especially of the roles of the mysterious MyTH4-FERM tandem, has been halted for over a decade. Here, for the first time, we provided high-quality live-cell images showing KCBP's cortical MT localizations in *Arabidopsis* epidermal pavement cells, hypocotyl cells, and trichome cells (*Figures 1, 2, Figure 1—figure supplement 2B, Figure 2—figure supplement 1; Videos 1–4*); thus, this work reveals the direct evidence linking the interphase KCBP function with trichome morphogenesis.

Furthermore, although KCBP has long been assumed to be part kinesin and part myosin, the direct evidence linking KCBP with the actin cytoskeleton is still lacking. The MyTH4 domain and the FERM domain are usually found in a tandem in myosins of the Myosin VII, X, and XV families, which differ substantially from the Myosin VIII and XI families found in the plant lineage. Myo VII contains a pair of MyTH4-FERM tandem domains, the second FERM domain of the *Drosophila* Myo VIIa binds to actin at high density (*Yang et al., 2009*). Unlike Myo VII, the MyTH-FERM tandem of Myo X, but not either of the isolated domains, was shown to bind to MTs (*Weber et al., 2004*; *Kerber and Cheney, 2011*) and also binds its cargo proteins, including β-integrins and the axonal guidance receptor DCC (*Zhang et al., 2004*; *Zhu et al., 2007*; *Wei et al., 2011*). Interestingly, in *Drosophila* myosin XV Sisyphus, the MyTH4 domain binds to MTs, whereas the FERM domain binds to various cargos including the MT-severing protein, Katanin p60 subunit (*Liu et al., 2008*). In the present study, by single-molecule imaging, we dissected the functional domain of the N-terminal tail and demonstrated that the NT plus the MyTH4 domain functions as a second MT-binding region besides the motor domain (*Figure 7A*), and the FERM domain physically binds to F-actin (*Figure 7B*). Thus, this work provides the first direct evidence linking the MT-based KCBP motor with the actin cytoskeleton, and further indicates that KCBP functions as a hub protein to mediate the interplay between MTs and actin filaments. Most importantly, our work solves the long-standing mystery of the function of the unique MyTH4-FERM domain in KCBP. We propose that the introduction of the MyTH4-FERM domain into the N-terminal tail region of a Kinesin-14 motor during evolution conferred the novel functions of KCBP: the NT and the MyTH4 domain act synergistically to bind strongly to MTs, but still maintain the MT-binding activity of the individual domains, and the FERM domain carries out the novel actin-binding activity, thus, orchestrating the interaction between MTs and the actin cytoskeleton.

Our speculation is supported by the genetic evidence that the KCBP containing the NT-MyTH4 truncation could not rescue the *kcbp-1* phenotype, but either the NT-truncated version of KCBP or the MyTH4-truncated version could fully complement the *kcbp-1* phenotype (*Figure 5B*).

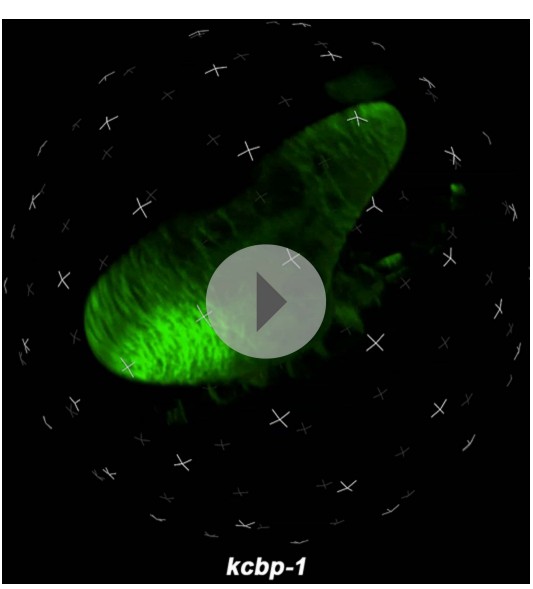

**Video 7.** The 3-D reconstructed cortical MT configuration in stage 3/4 *kcbp-1* trichomes.

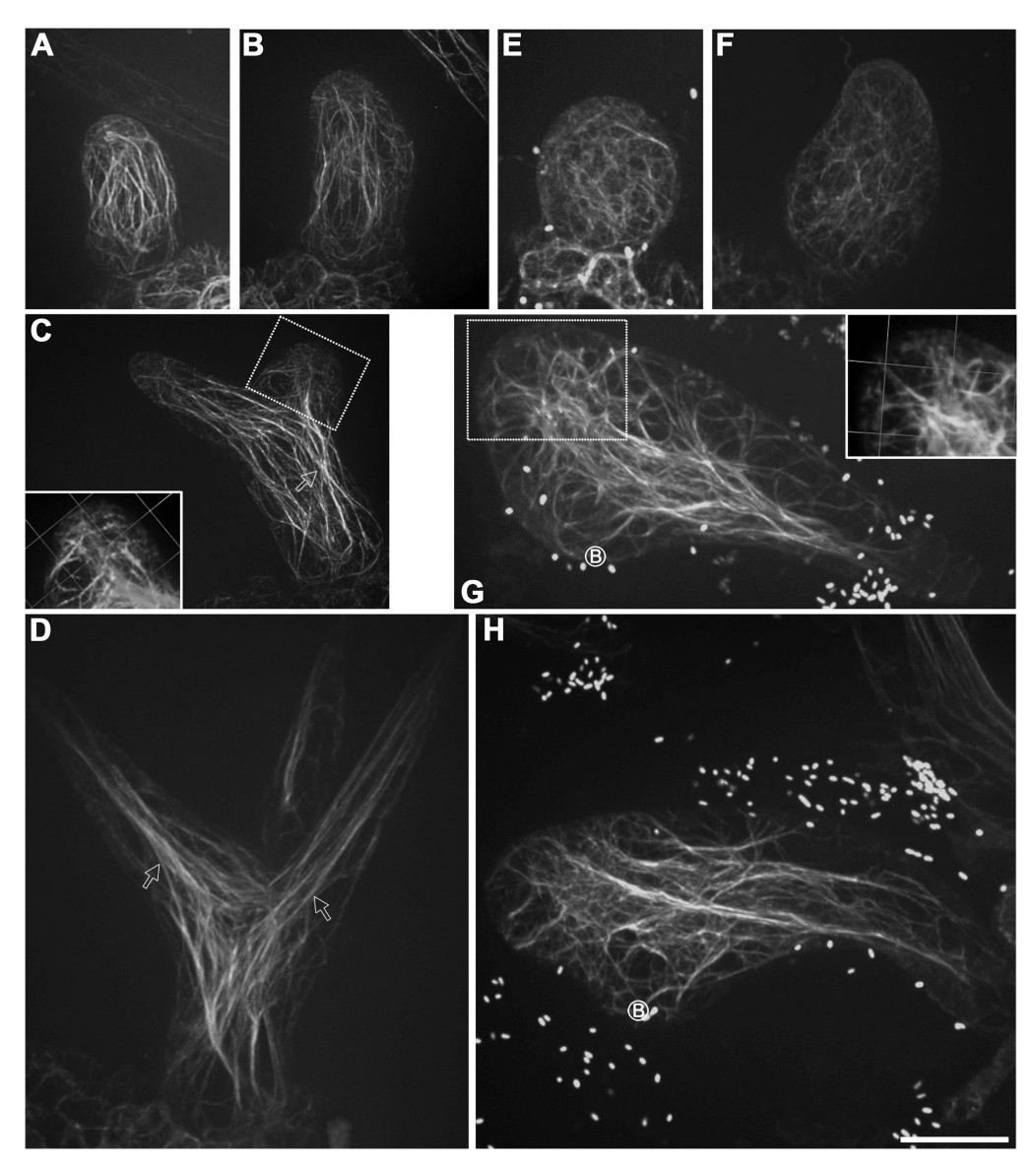

**Figure 4**. Aberrant organization of F-actin in *kcbp-1* trichomes. (**A–D**) Spatio-temporal organization of F-actin in wild-type trichomes during development. In stage 1/2 trichomes, a population of cytoplasmic actin cables align with the growth axis (**A**, **B**). In stage 3/4/5 trichomes, the cytoplasmic actin cables extend from the base to the branch tip, while a fine, cortical F-actin cap mainly in the transverse pattern is present in the tip region of elongating branches (**C**, **D**). The 3-D reconstructed image (the inset in **C**) generated from the area outlined by the dotted box highlights the F-actin. White open arrows highlight the parallel cytoplasmic actin cables. (**E–H**) Spatio-temporal organization of F-actin in *kcbp-1* trichomes during development. The stage 1/2 trichomes display random meshworks of thick actin bundles (**E**, **F**). At stage 3/4/5, the curly intertwined meshwork of thick actin bundles dominates inside trichomes, but parallel-aligned cytoplasmic actin cables and the fine F-actin cap at the branch apex as shown in wild-type are lost (**G**, **H**). The 3-D reconstructed image (the inset in **G**) generated from the area outlined by the dotted box highlights the disorganized F-actin in the tip region of the incipient primary branch. The circled B indicates the base of trichome stalks. The maximum z-projection of image stacks at 0.2-μm intervals was applied to all figures. Scale bars, 20 μm. One grid unit in the inset of (**D**, **E**) indicates 7.31 μm.

These findings indicate that the second MT-binding domain (NT+MyTH4) is essential for trichome morphogenesis, yet either the NT or the MyTH4 domain is fully functional, and compensate each other to bind to MTs. Surprisingly, either the FERM-truncated KCBP or the MyTH4-FERM-truncated KCBP

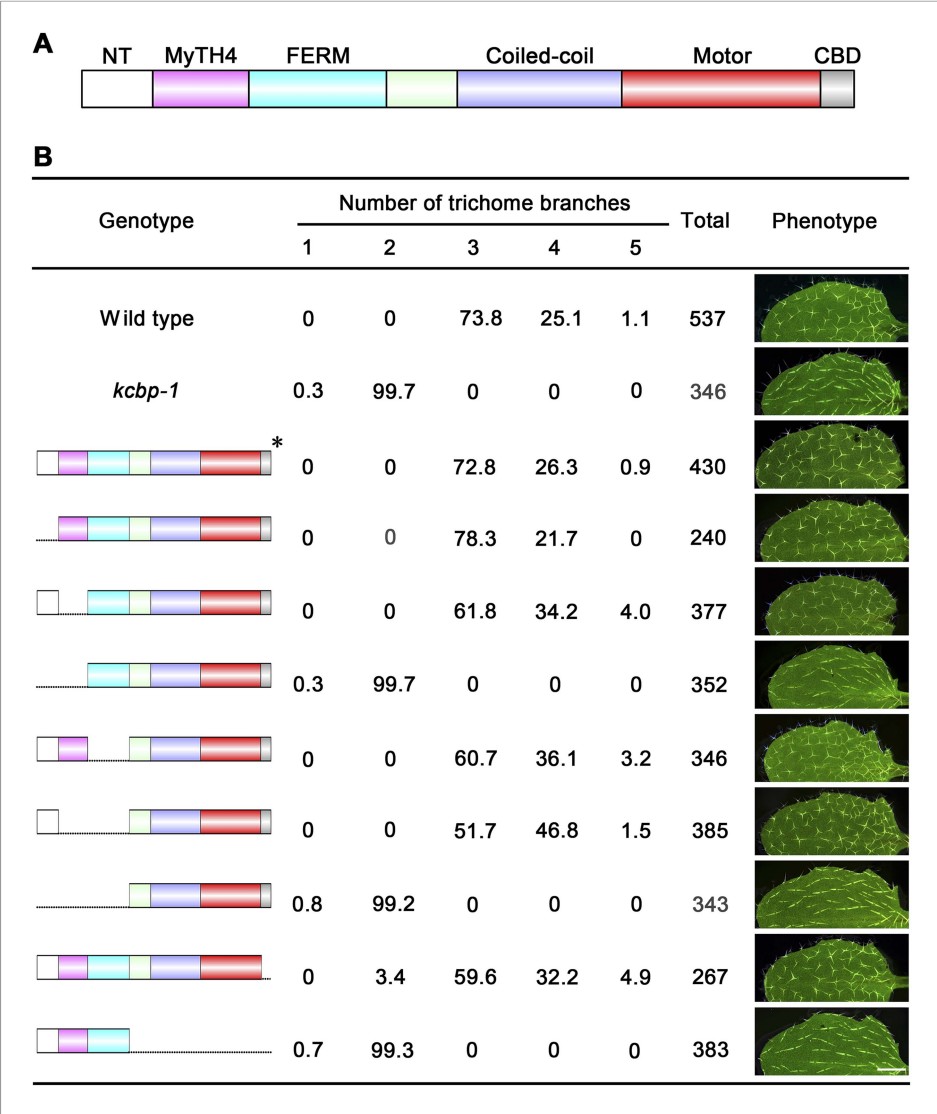

**Figure 5**. Genetic analyses on the role of individual domains of KCBP in trichome development. (**A**) Schematic diagram of the domain organization of KCBP. (**B**) Genetic complementation test using various truncated versions of KCBP. The genotype column shows the individual constructs containing various truncations used for the genetic complementation in the *kcbp-1* background. The phenotype column shows leaf trichomes in various transformants. Scale bars, 1 mm. The asterisk indicates the GFP-KCBP genomic fusion used in the complementation test.

could rescue the *kcbp-1* phenotype (*Figure 5B*). This unexpected finding indicates that the FERM domain is dispensable (*Figure 5B*), despite its actin-binding activity. However, our spatio-temporal observation revealed that loss-of-function mutation of *KCBP* results in defects in both MTs and F-actin (*Figures 3, 4*). Moreover, the interaction between KCBP and the actin cytoskeleton was genetically highlighted by the phenotype of plants expressing the dominant-negative rigor-KCBP construct, which show twisting, swollen trichome phenotype, similar to aberrant trichome morphology in a class of mutants of actin-related genes (*Figure 6*, *Figure 6—figure supplement 1*, *Figure 6—figure supplement 2*). Therefore, we proposed that KCBP indeed regulates actin dynamics, yet its actin-related function is redundant with other proteins in a same pathway. One explanation is that KCBP may function redundantly with KCH kinesins in the same kinesin-14 subfamily, which is extremely expanded in plants. KCH kinesins not only are minus-end-directed kinesins as with KCBP but bind to both MTs and F-actin, acting as dynamic linkers between MTs and actin filaments (*Preuss et al., 2004*; *Frey et al., 2009*; *Petrasek and Schwarzerova, 2009*; *Schneider and Persson, 2015*). So, further live-cell imaging

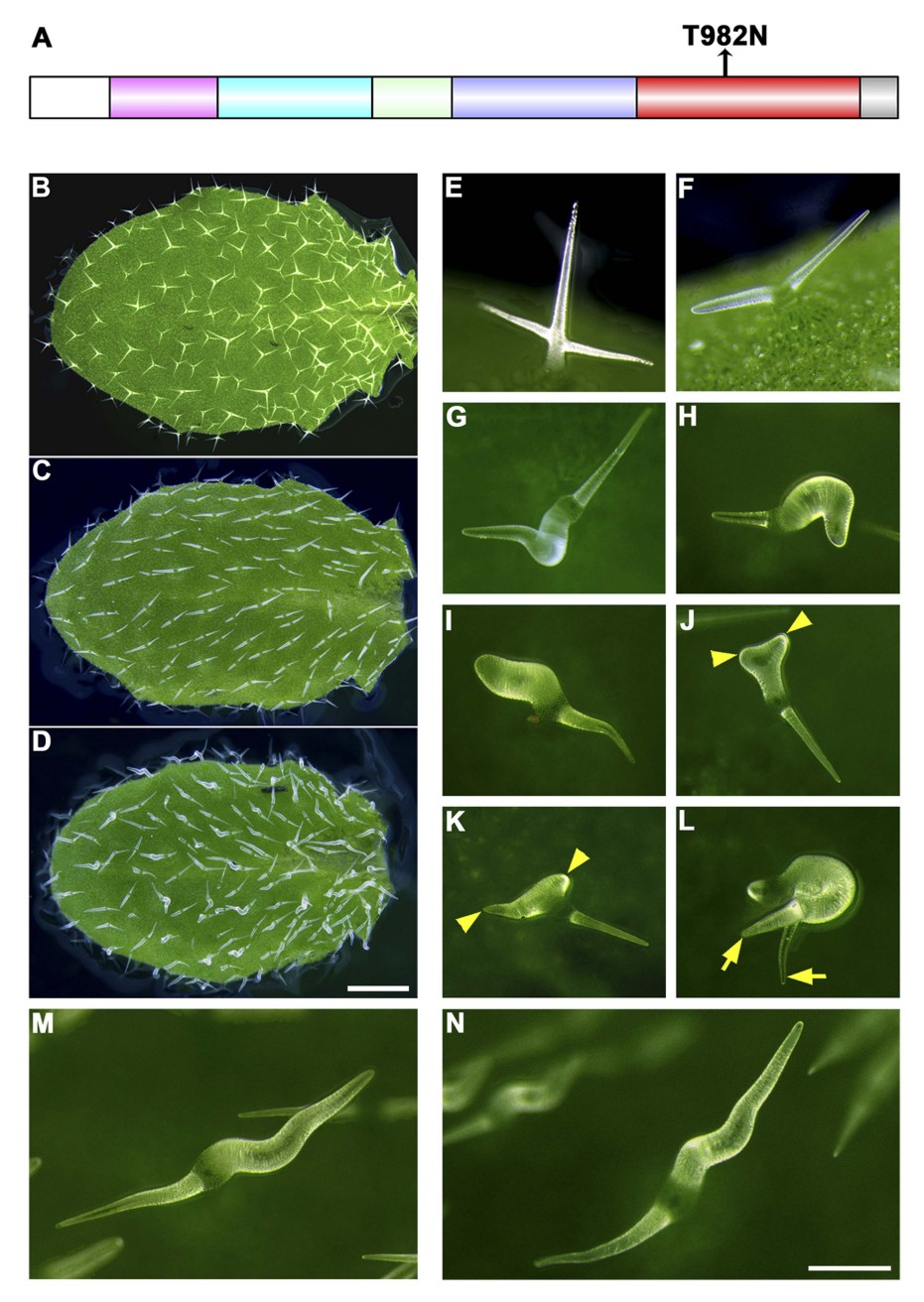

**Figure 6**. The trichome phenotype of rigor-KCBP transformants. (**A**) A point mutation was introduced into genomic *KCBP* including its native regulatory elements generate the rigor-KCBP with a threonine 982-to-asparagine substitution in its ATP-binding motif. The rigor-KCBP construct was introduced into the *kcbp-1* mutant to allow us to detect specific dominant-negative effects. (**B**, **E**) The typical wild-type trichomes contain a well-extended stalk and three/four branches. (**C**, **F**) Most *kcbp-1* trichomes contain an unextended stalk and two branches including one shortened branch, with a swollen and blunt tip. (**D**, **G–N**) Most rigor-KCBP trichomes contain two branches including one extremely twisted and swollen branch (**G–I**); a few of these trichomes show extremely elongated branches (**M**, **N**). A small portion of rigor-KCBP trichomes form three branches with one or two swollen and twisted branches (**J–L**). The arrowheads highlight unextended branches in a rigor-KCBP trichome. The arrows highlight two elongated branches with a fine tip, but in an irregular pattern, in a rigor-KCBP trichome. Scale bars are 1 mm in **B–D**, and 0.2 mm in **E–N**.

*Figure 6. continued on next page*

*Figure 6. Continued*

The following figure supplements are available for figure 6:

**Figure supplement 1**. Abnormal MT organization in rigor-KCBP trichomes.

**Figure supplement 2**. Abnormal organization of actin filaments in rigor-KCBP trichomes.

and genetic analyses are required to investigate the potential functional redundancy between KCHs and KCBP. Another possibility is that KCBP may be partially redundant with the actin nucleator ARP2/3 complex as well as the W/SRC complex. The second scenario is supported by previous findings that KCBP acts in the same genetic pathway with DIS1/ARP3, a member of the ARP2/3 complex, and IBT1/SCAR2, a subunit of the W/SRC complex (*Schwab et al., 2003*; *Zhang et al., 2005a*, *2005b*), and that ARP2/3 partially colocalizes with MTs in pavement cells and the W/SRC subunit ABIL3 binds to cortical MTs when overexpressed (*Jorgens et al., 2010*; *Zhang et al., 2013*). Interestingly, ARP2/3 and BRK1, a subunit of the W/SRC complex, were shown to specifically accumulate at the branch apex (*Dyachok et al., 2008*; *Yanagisawa et al., 2015*). Thus, most likely, KCBP functions redundantly with ARP2/3 and W/SRC complexes to organize the cortical F-actin at the branch apex.

## KCBP assembles the required cytoskeletal configuration for trichome branching, elongation, and tip sharpening

Trichome morphogenesis represents a unique growth mode comprising trichome branching and highly polarized diffuse growth of branch elongation accompanied by a tip-sharpening process (*Beilstein and Szymanski, 2004*; *Sambade et al., 2014*; *Yanagisawa et al., 2015*). However, the exact nature of the cytoskeletal organization that enables the unique growth mode has not yet been elucidated. Here, we provided live-cell images in high quality, showing the spatio-temporal dynamics and 3-D reconstruction of cytoskeletal organization during trichome morphogenesis. Our observations clearly showed that the transverse MT rings encircling elongating branches and the MT-depleted zone at the extreme apexes are required for trichome elongation and tip sharpening, consistent with previous reports (*Beilstein and Szymanski, 2004*; *Yanagisawa et al., 2015*). Moreover, our observations also revealed that parallel actin cables along the growth axis and the cortical F-actin cap are required for trichome elongation and tip sharpening. Several previous reports also documented the cortical F-actin cap near the branch tips of elongating trichomes, but its configuration remains a matter of debate because of the technical challenges of live-cell imaging of trichome tips. By immunostaining, despite the very dim signal, the cortical F-actin cap was detected as roughly showing a transverse pattern (*Zhang et al., 2005a*), but phalloidin staining showed a clear F-actin cap mainly in a longitudinal pattern (*Yanagisawa et al., 2015*). Here, we acquired high-quality images in live trichomes using ABD2-GFP labeling, and these clearly showed the F-actin cap mainly in a transverse pattern (*Figures 2, 4*; *Video 3*). We consider that live-cell imaging with ABD2-GFP gives a weaker signal, but should reflect the natural scenario. We also note that the fine F-actin cap at trichome branch tips is distinct from the actin fringe that is seen in a longitudinal pattern at the subapical domain of pollen tubes, which undergo tip growth (*Lovy-Wheeler et al., 2006*; *Cheung et al., 2010*; *Wu et al., 2010*; *Su et al., 2012*). Instead, the specific cytoskeletal configuration, especially for the MT-depleted zone and the cortical F-actin cap near the trichome branch tip, may be associated with the unique, polarized diffuse growth of trichome branch elongation accompanied by tip sharpening during trichome cell shaping.

Strikingly, in *kcbp-1* trichomes, these specific cytoskeletal systems are compromised or disrupted (*Figures 3, 4*), indicating the essential role of KCBP in assembly of the required cytoskeletal systems for trichome cell shape control. This notion is further supported by our important discovery that KCBP exhibits a cortical gradient along elongating branches and concentrates at the extreme apexes, where are colonized by the MT-depleted zone and the cortical F-actin cap (*Figure 2*). Our findings indicated that KCBP is required for establishing the transverse MT rings to promote branch elongation, and that, through the FERM domain, KCBP may dominantly bind to cortical F-actin. This binding occurs at higher density at the extreme apex, as suggested by previous findings (*Yang et al., 2009*), and plays a critical role in the assembly of the F-actin cap, which then facilitates the maintenance of the

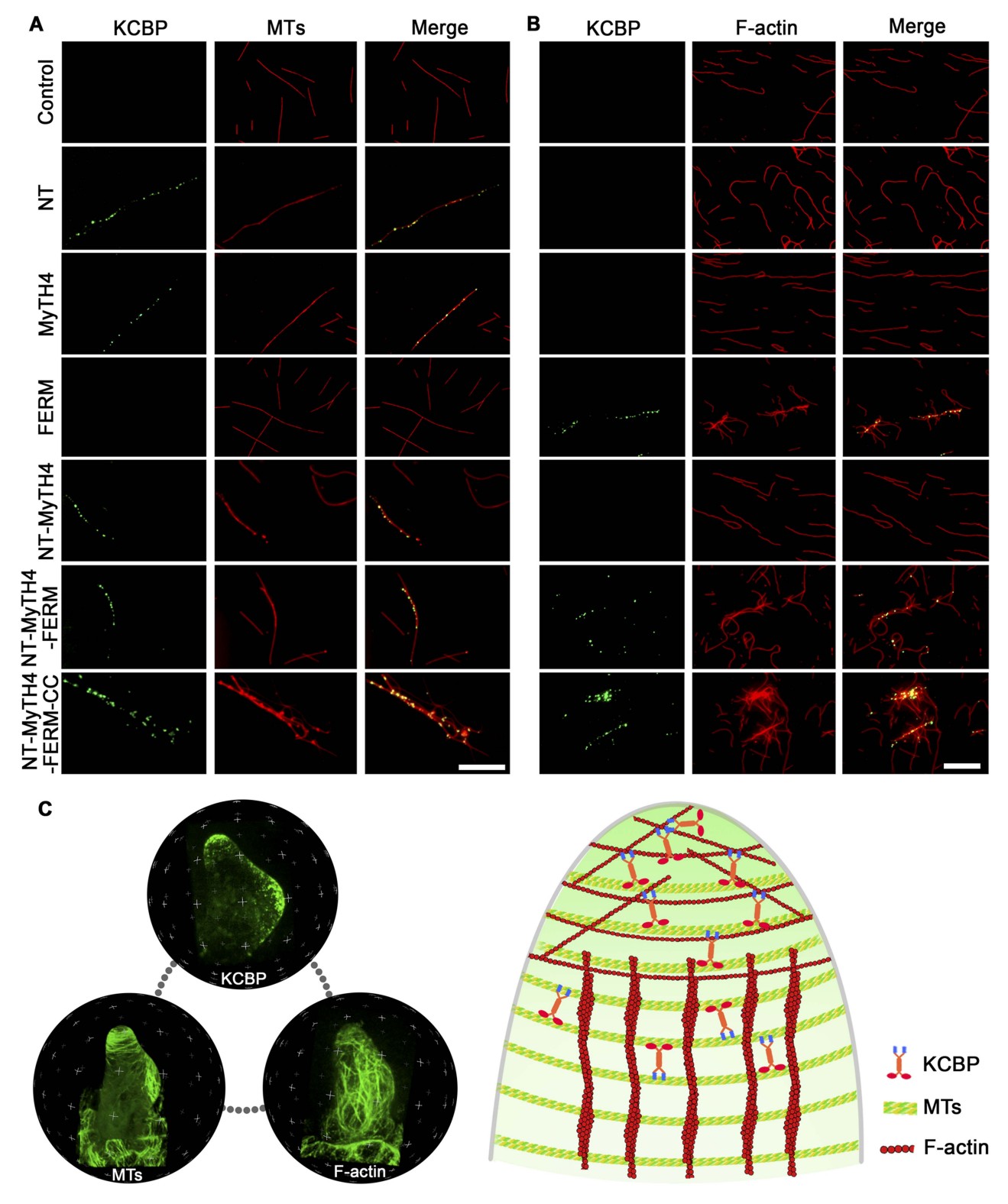

**Figure 7**. In vitro MT- and F-actin-binding activity of the motorless KCBP and a working model for KCBP during trichome morphogenesis. (**A**) The KCBP N-terminal tail containing the MyTH4 domain binds to MTs in vitro. Rhodamine-labeled MTs were incubated with GFP-NT, GFP-MyTH4, GFP-FERM, GFP-NT-MyTH4, GFP-NT-MyTH4-FERM, GFP-NT-MyTH4-FERM-CC recombinant proteins, or control GFP, respectively. GFP-NT-MyTH4-FERM-CC, GFP-NT-MyTH4-FERM, GFP-NT-MyTH4, GFP-NT, and GFP-MyTH4 exhibited a punctate pattern along MTs. Among them, GFP-NT-MyTH4-FERM-CC

*Figure 7. continued on next page*

*Figure 7. Continued*

promoted the formation of densely packed MT bundles. Scale bars, 10 μm. (**B**) The FERM domain binds to F-actin in vitro. Actin filaments were visualized in the presence of Alexa561-phalloidin. Alexa561-phalloidin labeled F-actin was incubated with GFP-NT, GFP-MyTH4, GFP-FERM, GFP-NT-MyTH4, GFP-NT-MyTH4-FERM, GFP-NT-MyTH4-FERM-CC recombinant proteins, or control GFP, respectively. GFP-NT-MyTH4-FERM-CC, GFP-NT-MyTH4-FERM, and GFP-FERM exhibited a punctate pattern along actin filaments. Among them, GFP-NT-MyTH4-FERM-CC promoted the formation of F-actin bundles. Scale bars, 5 μm. (**C**) A working model for KCBP during trichome cell shaping (right panel). The three spheres on the left panel show the 3-D reconstructions of the KCBP localization, the MT configuration, and the F-actin configuration in developing trichomes (at stage 2/3), respectively.

The following figure supplement is available for figure 7:

**Figure supplement 1**. Coomassie blue-stained SDS-PAGE gel of the purified GFP-KCBP recombinant proteins with various truncations.

MT-depleted zone to promote rapid elongation of trichome branches and tip sharpening (see model in *Figure 7C*). Recent work showed that ARP2/3 and BRK1, a subunit of the W/SRC complex, specifically accumulate at the branch apex, and plasma-membrane associated ARP2/3 and W/SRC complexes likely nucleate cortical F-actin at the branch apex (*Dyachok et al., 2008*; *Yanagisawa et al., 2015*). Therefore, most likely, KCBP functions in concert with the ARP2/3 complex as well as the W/SRC complex to assemble the specific cytoskeletal configuration near the branch tip region during trichome cell shaping. It would also be interesting to examine whether a local MT disassembly machinery occurs at the extreme branch apex to generate the MT-depleted zone, but an intriguing hypothesis is that KCBP might recruit the Katanin p60 subunit via its FERM domain, as suggested by the previous finding from the MyTH4-FERM-containing Sisyphus myosin motor (*Liu et al., 2008*), to locally disassemble MTs.

During trichome branching, we propose that once the branching program initiates, by a yet-unknown mechanism, the branching site becomes an isotropic zone with randomized sparse MT network; then, KCBP is required to rapidly colonize that area and organize the new transverse MT rings and the cortical F-actin cap to prime the initiation and subsequent elongation of the incipient branch. Previous studies showed that treatment with the MT stabilizing drug taxol can produce second branches in *kcbp* trichomes, albeit at the relatively low frequency of 5.5%, indicating that transient stabilization of MTs could compensate for the loss of KCBP activity (*Mathur and Chua, 2000*; *Buschmann et al., 2009*). Importantly, we revealed that KCBP exhibits non-processive movement by the live-cell, single-molecule (particle) motility assay (*Figure 1C,D*), and that the second MT-binding domain is essential, based on the genetic complementation tests (*Figure 5*); our data further support the bundling or crosslinking role of KCBP in the formation of the MT rings at the branching site. KCBP may further function in concert with AN1, a CtBP/BARS protein implicated in Golgi membrane trafficking and interacts with KCBP both physically and genetically (*Folkers et al., 2002*; *Kim et al., 2002*; *Minamisawa et al., 2011*), to integrate cytoskeletal dynamics and membrane trafficking to promote trichome branch initiation.

In conclusion, KCBP acts as a central player or a hub integrating the MT and actin cytoskeleton and further orchestrates their interplay to generate a specific cytoskeletal configuration that is required for trichome branching, elongation, and tip sharpening (see model in *Figure 7C*). Our findings reveal the in vivo function of the plant unique kinesin KCBP and further provide significant insights into molecular and cellular mechanisms that will close the knowledge gap between the cytoskeletal regulation and cell shape control, especially for the shaping of leaf trichome cells, which represents a unique growth mode comprising trichome branching, polarized branch elongation, and tip sharpening.

## Materials and methods

### Plant materials and growth conditions

*A. thaliana* ecotype Columbia (Col-0) was used as the genetic background in this study. Among the *zwichel* (*zwi*) alleles, the *zwi-3* allele and *zwi-w2* allele were frequently used in previous studies. The *zwi-3* allele in Columbia ecotype background, which is expected to produce a truncated ZWI protein (KCBP) at 522 amino acids position lacking the coiled-coil and motor domains, shows the typical, strong *zwichel* trichome phenotype (*Oppenheimer et al., 1997*; *Krishnakumar and Oppenheimer, 1999*). And the *zwi-w2* allele in RLD ecotype background shows weaker *zwichel* trichome phenotype,

containing a small portion (about 15.9%) of three-branched trichomes, because sequencing analysis revealed that although a C to T transition results in a stop codon at amino acid position 72, re-initiation of translation likely occurs using the in-frame AUG ~20 bp downstream from that mutation site as the start codon (*Folkers et al., 2002*). To find a strong or null *KCBP/ZWICHEL* allele in Columbia ecotype background, we searched the Salk collection in the *Arabidopsis* Biological Resource Center, and finally selected the accession of Salk_031704, which contains a T-DNA insertion in the third exon (22 exons in the *KCBP/ZWICHEL* gene in total) and shows typical, strong *zwi* trichome phenotype. Coincidently, the Salk_031704 strong allele was used in recent studies and was designated either as *kcbp-1* (*Humphrey et al., 2015*) or as *zwiA* (accession is N531704), which was ordered from the Nottingham *Arabidopsis* Stock Centre (*Buschmann et al., 2015*). The ABD2-GFP marker line is kindly provided by Prof. Shanjin Huang (Tsinghua University). The GFP-TUB6 marker line was described in our recent study (*Liu et al., 2014*), and the mCherry-TUB6 marker line was generated using the identical method, only replacing the GFP-encoding sequence with the mCherry-encoding sequence. Plasmid construction and generation of GFP-KCBP and rigor-KCBP lines can be found in the following parts, correspondingly. Various cross combinations (the GFP-KCBP line and the mCherry-TUB6 marker line; the *kcbp-1* mutant and the GFP-TUB6 marker line; the rigor-KCBP line and the GFP-TUB6 marker line; the *kcbp-1* mutant and the ABD2-GFP marker line; the rigor-KCBP line and the ABD2-GFP marker line) were performed to obtain corresponding materials to observe the dynamics of KCBP, MTs, and F-actin, respectively.

Plant growth conditions and transformation procedures were as described previously (*Liu et al., 2014*). The tiny first or second true leaves in the seedlings at 10-day-old stage were dissected and used for live-cell imaging of trichomes at various developmental stages.

## Plasmid construction for genetic complementation experiments

In general, the Phusion DNA polymerase with high-fidelity (New England Biolabs, Beverly, MA, United States) was used to amplify all the required gene products in this study, Fusion PCR was applied to get the GFP-KCBP fusion fragment and various constructs containing individual domain truncations (*Szewczyk et al., 2006*), and the gateway-based technology was applied to get final expression constructs, please refer to our recent study for detailed description (*Liu et al., 2014*).

To get the GFP-KCBP construct, the genomic fragment of the *KCBP* gene, including its coding sequence and the 761 bp upstream fragments from the translation initiation codon ATG, was amplified from the genomic DNA with primers of KCBPP-F and GA-KCBP-R. The amplified fragment was cloned into the pENTR/D-TOPO vector by a TOPO-based cloning strategy to get the Entry 1 clone according to manufacturer's instruction (Invitrogen, Carlsbad, CA, United States). Then, a VisGreen version (*Teerawanichpan et al., 2007*; *Liu et al., 2014*) of GFP tag was added to the NT of KCBP by the following manipulations. The promoter region of the *KCBP* gene was amplified with primers of KCBPP-F and KCBPP-R; the GFP-encoding sequence was amplified with primers of PGFP-F and GFP-R; the first part of *KCBP*-coding sequence (1–2050 bp) was amplified with primers of KCBP1X-F and KCBP1X-R. The above-mentioned three PCR fragments were further linked together by Fusion PCR using primers of KCBPP-F and KCBP1X-R. The resulting PCR fragment was subsequently cloned into the pENTR/D-TOPO vector to get the Entry 2 clone. Finally, both the Entry 1 vector and Entry 2 vector were digested by *Not* I and *Xba* I, and further ligation reaction was conducted between the gel-purified fragment containing the latter part of *KCBP* (1951–6023 bp) and the fragment containing the promoter, GFP, and the first part of *KCBP* genomic sequence, the resulting Entry 3 clone was delivered into pEarleyGate302 by recombination reaction to get the final GFP-KCBP construct.

A series of constructs containing various domain truncations were made by the following manipulations. To get the KCBP-ΔMyTH4 (KCBP lacking 117–275 amino acids) construct, the fragment containing promoter and 1–435 bp KCBP genomic sequence was amplified with primers of KCBPP-F and KCBP-ΔMyTH4-R, and the 992–6023 bp of the KCBP genomic sequence was amplified with primers of KCBP-ΔMyTH4-F and GA-KCBP-R, then the two PCR fragments were linked together by Fusion PCR using primers of KCBPP-F and GA-KCBP-R. Finally, the resulting PCR fragment was cloned into pENTR/D-TOPO vector and was then delivered into pEarleyGate302 to get the final KCBP-ΔMyTH4 construct. The same strategy was used to generate constructs of KCBP-ΔNT (KCBP lacking 2–121 amino acids), KCBP-ΔNT-MyTH4 (KCBP lacking 2–275 amino acids), KCBP-ΔFERM (KCBP lacking 275–497 amino acids), KCBP-ΔMyTH4-FERM (KCBP lacking 116–505 amino acids), KCBP-ΔNT-MyTH4-FERM (KCBP lacking 2–505 amino acids), KCBP-ΔCC-Motor-CBD (KCBP

lacking 532–1266 amino acids), and KCBP-ΔCBD (KCBP lacking 1210–1266 amino acids) primers can be found in the *Supplementary file 1*.

To get the rigor-KCBP construct, we introduced threonine (ACT) 982-to-asparagine (AAC) mutation by the PCR-based mutagenesis. In detail, we designed a pair of overlapping primers (KCBP-T982N-R and KCBP-T982N-F) with the desired nucleotide changes at the target site, and amplified the front half with primers of KCBP2X-F and KCBP-T982N-R, and that latter half with primers of KCBP-T982N-F and GA-KCBP-R, then the two fragments were linked together by Fusion PCR using primers of KCBP2X-F and GA-KCBP-R. The resulting PCR fragment was cloned into the pENTR/D-TOPO vector to get the Entry 4 clone. Finally, both the Entry 1 vector and the Entry 4 vector were digested by *Not* I and *Xba* I, and further ligation reaction was conducted between the gel-purified fragment containing the latter part of KCBP containing the threonine 982-to-asparagine mutation and the fragment containing the promoter and the first part of *KCBP* genomic sequence, the resulting Entry 5 clone was delivered into pEarleyGate302 to get the final rigor-KCBP construct.

## Genetic complementation and trichome phenotype identification

The *KCBP* (At5G65930) genomic fragment comprising its endogenous promoter and coding region was used for genetic complementation tests. Plasmid construction for the GFP-KCBP construct, the rigor-KCBP construct, and other constructs containing various domain truncations was all based on the above-mentioned *KCBP* fragment. The constructs were introduced into the *kcbp-1* mutant. The T3 homozygous transgenic lines were used to observe trichome phenotype under a fully automated Stereo Microscope (Leica M205 FA) with Leica Application Suite V4.2. The images of trichomes were a maximum z-projection of image series acquired by the Leica LAS Multifocus program.

## Spinning-disc confocal microscopy and motor motility analysis

Live-cell imaging was carried out under a spinning disk confocal microscope (UltraView VoX, Perkin Elmer, Beaconsfield, Buckinghamshire, UK) equipped with the Yokogawa Nipkow CSU-X1 spinning disk scanner, Hamamatsu EMCCD 9100-13, Nikon TiE inverted microscope with the Perfect Focus System as described previously (*Liu et al., 2014*). Acquired images were processed and analyzed using Volocity (Perkin Elmer), Image J (http://rsbweb.nih.gov/ij), MetaMorph (Molecular Devices, Sunnyvale, CA, United States).

The run-length distribution and the velocity distribution of GFP-KCBP were calculated in Origin software (OriginLab) by frequency counts. The mean values and 95% confidence interval were calculated in SAS (SAS Software), as described previously (*Kong et al., 2015*).

## Purification of GFP-tagged motor-less KCBP with various domain truncations

The full-length cDNA fragments of *KCBP* were amplified from the *Arabidopsis* cDNA with primers of KCBP1X-F and GA-KCBP-R, and the GFP-coding sequence amplified with primers of GFP-F and GFP-R, then the two PCR fragments were linked together by Fusion PCR using primers of GFP-F and GA-KCBP-R. The resulting PCR fragment was cloned into the pENTR/D-TOPO vector to get the Entry 6 clone. The cDNA fragments encoding polypeptide of GFP-NT-MyTH4-FERM-CC (1–749 amino acids), GFP-NT-MyTH4-FERM (1–614 amino acids), GFP-NT-MyTH4 (1–275 amino acids), GFP-MyTH4 (116–275 amino acids), GFP-FERM (276–503 amino acids), GFP-NT(1–115 amino acids), and GFP were generated based on the Entry 6 clone. The truncated fragments were digested with *Sal* I and *Not* I and were reconstructed into the pET-28a vector to get final expression constructs. Primers can be found in the *Supplementary file 1*.

Finally, the expression constructs were transformed into *Escherichia coli* strain Transetta (DE3, TransGen Biotech, Beijing, China) to induce expression. The recombinant proteins were purified using nickel-nitrilotriacetic acid (Ni-NTA) resin following procedures described by the manufacturer (Qiagen, Hilden, Germany). Fractions containing the protein were collected, combined, and dialyzed overnight against PEM buffer (80 mM PIPES, 1 mM EGTA (ethylene glycol tetraacetic acid), and 1 mM MgSO$_4$, pH 6.9). Protein concentration was determined by a Bio-Rad protein assay kit. Protein samples of 3 μg were analyzed by SDS-PAGE (Sodium Dodecyl Sulfate Polyacrylamide Gel Electropheresis).

## Single-molecule (particle) imaging assay

The purified porcine brain tubulin labeled with NHS-rhodamine was kindly provided from Prof. Tonglin Mao (China Agricultural University). Taxol-stabilized NHS-rhodamine MTs were incubated

with 1 µM GFP-NT, 1 µM GFP-MyTH4, 1 µM GFP-FERM, 1 µM GFP-NT-MyTH4, 1 µM GFP-NT-MyTH4-FERM, 1 µM GFP-NT-MyTH4-FERM-CC, and 1 µM control GFP, respectively, in PEM buffer at equal molar ratios for 30 min at room temperature, modified from a previous study (*Liu et al., 2013*). Fluorescent images of MTs and various GFP-KCBP truncated proteins were visualized using a Zeiss inverted fluorescence microscope (Axio Observer Z1) with a Zeiss Plan-Apochromat 100× oil immersion objective (NA = 1.4).

The rabbit skeletal muscle actins were provided by Prof. Shanjin Huang (Tsinghua University). F-actin (3 µM) was incubated with 1 µM GFP-NT, 1 µM GFP-MyTH4, 1 µM GFP-FERM, 1 µM GFP-NT-MyTH4, 1 µM GFP-NT-MyTH4-FERM, 1 µM GFP-NT-MyTH4-FERM-CC, and 1 µM control GFP at the indicated concentrations at room temperature for 30 min and then labeled with 3 µM Alexa561-phalloidin (Invitrogen). Actin filaments were subsequently diluted to a final concentration of 10 nM in fluorescence buffer (10 mM imidazole, pH 7.0, 50 mM KCl, 2 mM $MgCl_2$, 1 mM EGTA, 100 mM DTT (Dithiothreitol), 100 µg/ml Glucoxidase, 15 mg/ml Glucose, 20 µg/ml catalase, and 0.5% methylcellulose), modified from a previous study (*Wu et al., 2010*). The diluted samples were visualized using a Zeiss inverted fluorescence microscope (Axio Observer Z1) with a Zeiss Plan-Apochromat 100× oil immersion objective (NA = 1.4).

### Identification of T-DNA insertion in *kcbp-1* mutant

A standard PCR-based method was used to identify the T-DNA insertion in *kcbp-1* (*Humphrey et al., 2015*), as described by *Kong et al. (2015)*. Gene-specific primers (031704-LP and 031704-RP) were used to amplify an approximate 1000 bp DNA fragment in *KCBP*. The T-DNA insertion was detected using 031704-RP and the left-border primer (LBb1.3).

For RT-PCR analysis of transcription level of *KCBP* in the *kcbp-1* mutant, total RNA was extracted using Trizol reagent Invitrogen and was used for first strand cDNA synthesis by the SuperScript III First-Strand Synthesis System (Life Technologies, Carlsbad, CA, United States) with oligo (dT)18 primers. Then, the cDNA was used as a template for PCR reactions using primers shown in the primer list. *UBQ5* was used as control.

## Acknowledgements

The seeds of the *kcbp-1* mutant were kindly provided by Bo Liu from University of California at Davis, and we also appreciate his valuable comments on the work. We are grateful to Shanjin Huang from Tsinghua University for providing the seeds of ABD2-GFP marker line and kind help with in vitro actin-binding assay. We also thank Tonglin Mao from China Agricultural University for kind help with in vitro microtubule-binding assay. We greatly acknowledge Yao Wu and Lei Su from the Institute of Microbiology, Chinese Academy of Sciences, for providing technical assistance and training service. We also appreciate the technical support of the UltraView Vox system from Lei Jiao, Meng Lai, and Colin Hornby from Perkin Elmer. This study was supported by the National Science Foundation of China under Grant # 31171294, by the start-up fund of 'One Hundred Talents' program of the Chinese Academy of Sciences and by the grants from the State Key Laboratory of Plant Genomics (Grant # 2015B0129-02).

## Additional information

### Funding

| Funder | Grant reference | Author |
| --- | --- | --- |
| National Natural Science Foundation of China (NSFC) | 31171294 | Zhaosheng Kong |
| Chinese Academy of Sciences (CAS) | Startup funding of Hundred Talents Program | Zhaosheng Kong |
| Chinese Academy of Sciences (CAS) | Grant # 2015B0129-02 of the State Key Laboratory of Plant Genomics | Zhaosheng Kong |

The funders had no role in study design, data collection and interpretation, or the decision to submit the work for publication.

## Author contributions

JT, LH, Acquisition of data, Analysis and interpretation of data; ZF, GW, Analysis and interpretation of data, Contributed unpublished essential data or reagents; WL, YM, YY, Acquisition of data, Contributed unpublished essential data or reagents; ZK, Conception and design, Analysis and interpretation of data, Drafting or revising the article

## Additional files

### Supplementary file

• Supplementary file 1. List of primer sequences used in this study.

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
