## [Decision Letter]

Thank you for submitting your work entitled “Orchestration of Microtubules and the Actin Cytoskeleton in Trichome Cell Shape Determination by a Plant Unique Kinesin” for peer review at *eLife*. Your submission has been favorably evaluated by Detlef Weigel (Senior editor) and two reviewers, one of whom, Sheila McCormick, is a member of our Board of Reviewing Editors.

The reviewers have discussed the reviews with one another and the Reviewing editor has drafted this decision to help you prepare a revised submission.

Trichomes are an interesting model for cell differentiation. Previous analyses of trichome phenotypes caused by mutations in either microtubules or actin gave contrasting phenotypes, i.e. those lacking microtubules gave fewer branches and those lacking actin gave stubby branches. A publication in 1997 indicated that plants that lacked expression of KCBP (for Kinesin-like Calmodulin-Binding Protein, i.e. mutants of *Zwichel*) had a trichome phenotype that was suggestive of both a microtubule (MT) and actin problem, but no particular follow up with respect to the possible bridging role between MTs and actin was ever published. Here the authors study KCBP, which contains numerous domains. They show that two of the novel domains near the N-terminus bind microtubules (the MyTH4 domain) or actin (the FERM domain), and because of this they suggest that KCBP might provide the link (hub protein) needed to coordinate microtubules and actin during trichome development. They use fluorescent-tag imaging to show where this protein is localized in trichomes, i.e. in a gradient and concentrated at branching sites and tips of elongating branches, where there are not MTs but where there is cortical F-actin. They then use a series of deletion constructs to determine the roles of the different domains of the constructs, by introducing them into the kcbp mutant background - from these studies they eliminate a role for the calmodulin-binding domain, but surprisingly, also eliminate a role for the FERM domain. That is, it is mostly the N-terminus and the MyTH4 domain that are needed to complement the mutant phenotype.

Essential revisions:

1) What is the ‘unique tip-directed diffuse growth during trichome cell morphogenesis’ that the authors talk about? Where have the authors come across this unique growth? How is it possible to achieve this kind of growth? Trichomes in arabidopsis grow by diffuse growth which allows their tip region to be extended. Previous published descriptions of trichome development mention outgrowth, extension growth and branching. There is no tip-directed growth as observed in tip-growing pollen tubes or root hairs. Since they use their observations of a transverse cortical F-actin cap to propose association with the unique tip-directed diffuse growth, and because other conclusions and the model presented rely heavily on this supposed unique growth pattern, the authors must provide clear evidence that such a growth pattern actually exists.

2) The tip of a developing trichome is one of the oldest parts of the trichome cell. The region produces autofluorescence and the images provided do not distinguish autofluorescence and curvature-induced impression of a green signal from actual fluorescence of their probes. Perhaps a control should be included to convince.

3) It is not clear why they don't use the name zwichel, which is the name previously used for this gene. They state that they use the *kcbp-1* (SALK_031704) mutant for their studies. Which allele of *zwi*? Please provide a clear reference for this and discuss. Are the observations true for other alleles of *zwi*? If this has been checked it would strengthen their observations and conclusions. If not checked then there is a concern about the various zones of depletion and suggested interactions proposed by the authors.

4) It was confusing as to why a construct missing both MyTH4 and FERM were needed to “not rescue” the phenotype - i.e. lacking the ability to bind to one or the other binding partner of an important bridging protein should both give the “not rescue” answer, no? This needs more discussion/explanation. They do make a point mutation in the motor region (so-called rigor mutation) that they use to argue for a connection with actin. Can the argument about redundancy in the discussion be strengthened?

[Editors' note: further revisions were requested prior to acceptance, as described below.]

Thank you for resubmitting your work entitled “Orchestration of Microtubules and the Actin Cytoskeleton in Trichome Cell Shape Determination by a Plant Unique Kinesin” for further consideration at *eLife*. Your revised article has been favorably evaluated by Detlef Weigel (Senior editor) and a Reviewing editor. The manuscript has been improved but there are some remaining issues that need to be addressed before acceptance, as outlined below:

1) You have not sufficiently made it clear that *zwichel* and *KCBP* are the same gene, i.e. the only mention of the *At* identifer of this gene is in the methods, and the statement, “since *KCBP* was cloned and linked to the trichome phenotype of the *zwichel* mutation” is misleading. The term “linked to” might imply a genetic linkage. Furthermore, the discussion (in the response to reviewers) about the other *zwi* alleles and their issues should be included in the manuscript, so it is clear to that reader that this gene is also called *zwichel* (was it first called *zwichel* or first called *KCBP*?), that other alleles exist, but that they have problems, so a null is better, etc.

2) As mentioned before, *eLife* does not have page or word limits, so it is unclear why you chose to include a clarifying figure and video in the response to the reviewer comment about a needed control but did not insert this information into the manuscript. i.e. You state in the response letter:

“Finally, as suggested, we used GCP2 (30), a component of the MT nucleation complex, as a control. Interestingly, we observed that, similar to the MTs, GCP2 shows a tip-directed gradient, but with a GCP2-depleted zone at the extreme apexes of elongating branches (please see the Figure 8 and the Video 8).”

3) Figure 9 should also be included as a supplemental file, and it and its legend could be more helpful - i.e. the location of the primers are not shown - do you expect the reader to go to the list of primers in the table and do their own search to discover this information? Is the reader required to go look at [19] to find out which primers you might have used? Did you use the same primers? This is relevant since you claim in the response to reviewers that *kcbp-1* is better than *zwi-w2*, but *zwi-w2* has a peptide of 72 amino acids, and since the T-DNA in *kcbp-1* is in the 3rd exon, the possibility also exists that it might have a small N-terminal peptide, no?

---

## [Author Response]

*1) What is the ‘unique tip-directed diffuse growth during trichome cell morphogenesis’ that the authors talk about? Where have the authors come across this unique growth? How is it possible to achieve this kind of growth? Trichomes in arabidopsis grow by diffuse growth which allows their tip region to be extended. Previous published descriptions of trichome development mention outgrowth, extension growth and branching. There is no tip-directed growth as observed in tip-growing pollen tubes or root hairs. Since they use their observations of a transverse cortical F-actin cap to propose association with the unique tip-directed diffuse growth, and because other conclusions and the model presented rely heavily on this supposed unique growth pattern, the authors must provide clear evidence that such a growth pattern actually exists*.

We are sorry for the confusion caused by the expression of “tip-directed diffuse growth” in our manuscript. We totally agree with your opinion that “There is no tip-directed growth as observed in tip-growing pollen tubes or root hairs”. By tracking the beads that had been placed on the surface of elongating trichome branches, the extension of trichome branches had been established to occur along the whole cell surface, clearly indicating that trichome branches grow in the mode of diffuse growth. Interestingly, the experiments meanwhile showed that diffuse-growing trichome branches elongate in a polarized manner, the distal regions of the branch grows at significantly faster rates than the branch base (please see Figure 4 in [46]; please also see Figures 1c and 1d in [58]). Therefore, the growth mode of trichome branches is definitely diffuse growth, but in a highly polarized manner towards the tip. Thus, this growth mode is often referred to as “tip-biased diffuse growth” or “highly polarized diffuse growth” in literatures (53; 43; 58).

In our manuscript, we only wanted to use the “tip-directed diffuse growth” to describe the unique growth mode, which combines the highly-polarized diffuse growth with the tip sharpening process, during trichome branch elongation. In addition, we didn’t mean that we found the tip growth based on the observation of the transverse cortical actin cap. Instead, we speculated that the transverse cortical actin cap and the microtubule (MT)-depleted zone are tightly associated with the tip sharpening.

To avoid the confusion and to precisely describe the unique growth mode of trichome morphogenesis, we replaced the “tip-directed diffuse growth” with the “polarized diffuse growth” in the revised version. We also corrected the confusing descriptions through the manuscript, and emphasized that KCBP acts as a central player or a hub protein to assemble the required cytoskeletal configuration for trichome branching and highly polarized branch elongation accompanied by tip sharpening.

*2) The tip of a developing trichome is one of the oldest parts of the trichome cell. The region produces autofluorescence and the images provided do not distinguish autofluorescence and curvature-induced impression of a green signal from actual fluorescence of their probes. Perhaps a control should be included to convince*.

Thanks for your valuable suggestions. First, as mentioned earlier, the tip regions of the elongating branch grows at significantly faster rates than the branch base (46, 58). The wall thickness measurements by transmission electron microscopy (TEM) showed that there is a tip-to-base wall thickness gradient in elongating trichomes and the tip region is the thinnest part; another signal intensity analysis using propidium iodide further confirmed the wall thickness gradient (please see Figures 2g, 2h and 2i in [58]). Thus, in elongating trichomes, the cell wall autofluorescence at the tip should be weaker than that at the base, and should not affect the observation of KCBP localization pattern. Otherwise, we couldn’t detect the MT-deplete zone at the extreme apexes of elongating branches in Figure 2.

Second, to remove your concern that the accumulation of KCBP at the tip region might be the curvature-induced impression, we displayed the image serials that were used to make the Z-projections and 3-D reconstructions. These original images also clearly show that KCBP is concentrated at tip regions where show very weak signal of MTs (please see the new Figure 2—figure supplement 2). Finally, as suggested, we used GCP2 (30), a component of the MT nucleation complex, as a control. Interestingly, we observed that, similar to the MTs, GCP2 shows a tip-directed gradient, but with a GCP2-depleted zone at the extreme apexes of elongating branches (please see the Figure 8 and the Video 8).

Author response image 1.Localization of GCP2 in stage 2/3 wild-type trichomes. (A) The Z-projection image, which was acquired from a high-resolution stack of 46 planes at 0.2 μm intervals, shows a tip-oriented cortical gradient of GCP2-3×GFP particles in the elongating main stem in a stage 2/3 trichome. Absence of GCP2-3×GFP was observed at the extreme apexes of the main stems. Scale bars, 10 μm.(B) Absence of GCP2-3×GFP at the extreme apexes of the main stems is highlighted by the 3-D reconstruction of stage 2/3 trichomes. One grid unit indicates 5.63 μm. (C) The GCP2-3×GFP images, which were used to make the Z-projection in (A), were sequentially illustrated at 0.4 μm intervals. Absence of GCP2-3×GFP was observed at extreme apexes of elongating branches in stage 2/3 trichome. Scale bars, 10 μm.**DOI:**
http://dx.doi.org/

Author response video 1.The spatial distribution of GCP2-3×GFP is highlighted by 3-D reconstitution in a stage 2/3 wild-type trichome.**DOI:**
http://dx.doi.org/10.7554/eLife.09351.03110.7554/eLife.09351.031

*3) It is not clear why they don't use the name zwichel, which is the name previously used for this gene. They state that they use the kcbp-1 (SALK_031704) mutant for their studies. Which allele of zwi? Please provide a clear reference for this and discuss. Are the observations true for other alleles of zwi? If this has been checked it would strengthen their observations and conclusions. If not checked then there is a concern about the various zones of depletion and suggested interactions proposed by the authors*.

We understand your concern. Among the KCBP mutants of the *zwichel* (*zwi*) serial, the *zwi-3* mutant and *zwi-w2* mutant were frequently used in previous reports. The *zwi-3* mutant still produces a truncated ZWI protein (KCBP) lacking the coiled-coil and motor domains (38; 26); and the *zwi-w2* mutant still contains a peptide fragment of the first 72 amino acids of the KCBP protein (16). Although both *zwi-3* and *zwi-w2* are loss of function alleles, we had the same concern as you proposed, i.e., the remaining truncated version of KCBP may interact with other partners and interfere with the complementation effects. So the best choice is using a null allele for the genetic complementation tests. The *kcbp-**1* (Salk_031704) mutant allele, which shows the same trichome phenotype as with the loss-of-function alleles of *zwi-3* and the *zwi-w2*, is the best candidate because it contains a T-DNA insertion in the third exon, and was confirmed to be a null allele (please see Figure 3 in [19]). The homozygous *kcbp-1* lines, which were identified by detecting the T-DNA insertion (please see the Figure 9), were used to be the null KCBP background in our complementation tests, and the primers were already listed in the primer list in the previous version. We provided the reference when the *kcbp-1* firstly appeared in the Results part in the revised version.

Author response image 2.Genetic identification of the *kcbp-1* mutant. (A) Gene structure of the *KCBP* gene. Black rectangles indicate exons, gray rectangles indicate the 5’ and 3’ untranslated regions, and thick lines indicate introns. The arrow indicates the location of the T-DNA insertion in the Salk_031704 line. (B) PCR analysis of the T-DNA insertion. LP and RP indicate a pair of *KCBP* specific primers. BP indicates the T-DNA border primer.**DOI:**
http://dx.doi.org/10.7554/eLife.09351.032

4) It was confusing as to why a construct missing both MyTH4 and FERM were needed to “not rescue” the phenotype - i.e. lacking the ability to bind to one or the other binding partner of an important bridging protein should both give the “not rescue” answer, no? This needs more discussion/explanation. They do make a point mutation in the motor region (so-called rigor mutation) that they use to argue for a connection with actin. Can the argument about redundancy in the discussion be strengthened?

Thanks a lot for your valuable suggestions. We understand your confusion, so we have provided a more clear explanation, and the discussion about redundancy has also been strengthened, in the Discussion section of the revised version.

[Editors' note: further revisions were requested prior to acceptance, as described below.]

*1) You have not sufficiently made it clear that zwichel and KCBP are the same gene, i.e. the only mention of the At identifer of this gene is in the methods, and the statement, “since KCBP was cloned and linked to the trichome phenotype of the zwichel mutation” is misleading. The term “linked to” might imply a genetic linkage. Furthermore, the discussion (in the response to reviewers) about the other zwi alleles and their issues should be included in the manuscript, so it is clear to that reader that this gene is also called zwichel (was it first called zwichel or first called KCBP?), that other alleles exist, but that they have problems, so a null is better, etc*.

Sorry for the confusion. We have provided essential background information, which shows that *ZWICHEL* and *KCBP* are the same gene, but initially were characterized *via* different approaches by different groups, respectively. KCBP was firstly identified as a novel kinesin containing a calmodulin-binding domain by biochemical approaches using biotinylated calmodulin as a probe in 1996 (Reddy et al., 1996). The *zwichel* mutants were firstly characterized to show the typical *zwichel* trichome phenotype in 1994 (18), and then the *ZWICHEL* gene was cloned and found to encode the KCBP protein in 1997 (38). Please see paragraph 3 of the Introduction.

The information about *zwi* alleles has also been included in the revised manuscript. As mentioned earlier, among the *zwichel* (*zwi*) alleles, the *zwi-3* allele and *zwi-w2* allele were frequently used in previous studies. The *zwi-3* allele in Columbia ecotype background, which is expected to produce a truncated ZWI protein (KCBP) of 522 amino acids lacking the coiled-coil and motor domains, shows the typical, strong *zwichel* trichome phenotype (38; 26). And the *zwi-w2* allele in RLD ecotype background shows weaker *zwichel* trichome phenotype, containing a small portion (about 15.9 %) of three-branched trichomes, because sequence analysis revealed that although a C to T transition results in a stop codon at amino acid position 72, re-initiation of translation likely occurs using the in-frame AUG ∼20 bp downstream from that mutation site as the start codon (16). To find a strong or null *KCBP/ZWICHEL* allele in Columbia ecotype background, we searched the Salk collection in the Arabidopsis Biological Resource Center, and finally selected the accession of Salk_031704, which contains a T-DNA insertion in the third exon (22 exons in the *KCBP/ZWICHEL* gene) and shows typical, strong *zwichel* trichome phenotype. Coincidently, the Salk_031704 strong allele was used in recent studies, and was designated either as *kcbp-1* (19) or as *zwiA* (accession is N531704; http://arabidopsis.info/StockInfo?NASC_id=531704), which was ordered from the Nottingham Arabidopsis Stock Centre (7). Please see first paragraph of the Materials and methods section.

Please also find the more detailed information in the response to the Question 3 below.

*2) As mentioned before,* eLife *does not have page or word limits, so it is unclear why you chose to include a clarifying figure and video in the response to the reviewer comment about a needed control but did not insert this information into the manuscript. i.e. You state in the response letter*:

*“Finally, as suggested, we used GCP2 (*[30]*), a component of the MT nucleation complex, as a control. Interestingly, we observed that, similar to the MTs, GCP2 shows a tip-directed gradient, but with a GCP2-depleted zone at the extreme apexes of elongating branches (please see the Figure 8 and the Video 8**)*.*”*

Thanks a lot. We have included the original Figure 8 and the Video 8 in the revised manuscript. Please see the new Figure 2—figure supplement 3 and the new Video 5. Legends are also included.

*3) Figure 9 should also be included as a supplemental file, and it and its legend could be more helpful - i.e. the location of the primers are not shown - do you expect the reader to go to the list of primers in the table and do their own search to discover this information? Is the reader required to go look at*
[19]
*to find out which primers you might have used? Did you use the same primers? This is relevant since you claim in the response to reviewers that kcbp-1 is better than zwi-w2, but zwi-w2 has a peptide of 72 amino acids, and since the T-DNA in kcbp-1 is in the 3rd exon, the possibility also exists that it might have a small N-terminal peptide, no?*

Many thanks for your very careful examination and valuable suggestions in improving the manuscript. The Figure 9 has been included as the new Figure 1—figure supplement 1. Furthermore, we performed detailed genetic identification of the *kcbp-1/zwiA* allele according to your suggestion. Sequencing analysis revealed that the T-DNA insertion is in the third exon and located at the position of 957 bp downstream the ATP start codon of the *KCBP* gene (please see the new Figure 1—figure supplement 1, and its legend), and introduces a premature stop codon 61 bp downstream the insertion site by frame shift. As expected, we couldn’t detect the transcripts of the *KCBP* after the T-DNA insertion, but we indeed detected weaker transcription of partial *KCBP* in front of the insertion site (please see the new Figure 1—figure supplement 1). So, the *kcbp-1/zwiA* mutant is expected to produce a short peptide of 284 amino acids containing the first 263 amino acids of KCBP at a lower level. Nevertheless, the *kcbp-1/zwiA* allele is still the best choice for us because it shows strong, typical *zwichel* trichome phenotype, and is in Columbia ecotype background and publicly available, thus it is frequently used in recent studies (7; 19). In addition, we don’t think that the N-terminal 263 amino acids residual in the *kcbp-1/zwiA* mutant would affect the complementation tests because most of the constructs (6/9) could rescue the typical *zwichel* trichome defects (please see Figure 5). Actually, the *zwiA* had also been successfully used for the genetic complementation in the most recent study (7).